# Efficient Off-Policy Evaluation with Safety Constraint for Reinforcement Learning

**Claire Chen**[*]
School of Arts and Science
University of Virginia
`clairechen@email.virginia.edu`

**Shuze Daniel Liu**[*]
Department of Computer Science
University of Virginia
`shuzeliu@virginia.edu`

**Shangtong Zhang**
Department of Computer Science
University of Virginia
`shangtong@virginia.edu`

## Abstract

In reinforcement learning, classic on-policy evaluation methods often suffer from high variance and require massive online data to attain the desired accuracy. Previous studies attempt to reduce evaluation variance by searching for or designing proper behavior policies to collect data. However, these approaches ignore the safety of such behavior policies—the designed behavior policies have no safety guarantee and may lead to severe damage during online executions. In this paper, to address the challenge of reducing variance while ensuring safety simultaneously, we propose an optimal variance-minimizing behavior policy under safety constraints. Theoretically, while ensuring safety constraints, our evaluation method is unbiased and has lower variance than on-policy evaluation. Empirically, our method is the only existing method to achieve both substantial variance reduction and safety constraint satisfaction. Furthermore, we show our method is even superior to previous methods in both variance reduction and execution safety.

## 1 Introduction

Recently, reinforcement learning (RL, Sutton and Barto (2018)) has shown exceptional success in various sequential decision-making problems. For instance, the applications of RL algorithms have reduced energy consumption for Google's data center's cooling by $40\%$ (Chervonyi et al., 2022), solved Olympiad-level geometry problems (Trinh et al., 2024), and designed general-purpose data center CPUs (Mirhoseini et al., 2021). In RL applications, *policy evaluation* enables RL practitioners to estimate the performance of a policy before committing to its full deployment. In policy evaluation, conventional wisdom uses the on-policy method, in which a policy (i.e., the target policy) is evaluated by directly executing itself. However, this straightforward approach is crude since using the target policy itself as the data-collecting policy (i.e., the behavior policy) is proved to be suboptimal (Liu and Zhang, 2024; Liu et al., 2024a;c), resulting in evaluations with potentially high variance. Thus, the on-policy evaluation method may require massive online data to achieve the desired accuracy.

Unfortunately, collecting massive online data in real-world interaction can be both expensive and slow (Li, 2019; Zhang, 2023). In Google's data center's cooling system, each interaction step in the actual deployment takes 5 minutes (Chervonyi et al., 2022). Thus, the evaluation of a policy requiring millions of steps is prohibitively expensive. To reduce the reliance on costly online data collection, offline RL has been introduced as a possible solution. However, mismatches between the offline data distribution and the distribution induced by the target policy frequently arise, resulting in bias that is both uncontrolled and difficult to eliminate (Jiang and Li, 2016; Farahmand and Szepesvári, 2011; Marivate, 2015). As a result, both online and offline RL practitioners still depend heavily on online policy evaluation techniques (Kalashnikov et al., 2018; Vinyals et al., 2019).

---

[*]Equal Contribution.

To improve the online sample efficiency for policy evaluation, existing methods propose to reduce the evaluation variance by searching for or designing proper behavior policies (Hanna et al., 2017; Zhong et al., 2022; Liu and Zhang, 2024). However, all their approaches ignore a critical issue: *safety*. In many real-world applications, neglecting safety in policy execution can result in serious consequences. For example, in Google's data center cooling system, a behavior policy that is tailored for variance reduction without considering safety constraints may unpredictably overheat the system, causing equipment damages or service disruptions. Therefore, besides reducing evaluation variance, it is crucial to guarantee execution safety.

In this paper, we address the challenge of reducing variance while ensuring safety simultaneously. We make the following contributions:

1. We propose an optimal variance-minimizing behavior policy under safety constraints.

2. Theoretically, we show that our method gives an unbiased estimation. In addition to strictly satisfying the safety constraints, our method is proven to attain lower variance than the classic on-policy evaluation method.

3. Empirically, we show that our method is the only existing method to achieve both substantial variance reduction and constraint satisfaction. Moreover, it is even superior to previous methods in both variance reduction and execution safety.

## 2 RELATED WORK

**Safe RL.** Safety in reinforcement learning, often framed as safe RL (García and Fernández, 2015), has been an active research topic recently. Many recent works focus on safety in policy exploration and optimization (Brunke et al., 2022). For safe exploration, Moldovan and Abbeel (2012) present a method for ensuring safe exploration by keeping the agent within a predefined set of safe states during its learning process. However, their method is a model-based approach, requiring an explicit approximation of the transition function, which introduces challenges common to model learning, such as compounding errors and the need for accurate model dynamics (Sutton, 1990; Sutton et al., 2008; Deisenroth and Rasmussen, 2011; Chua et al., 2018). *In contrast, our approach does not rely on approximating the transition function (i.e., model-free), since parameters can be estimated by off-the-shelf offline policy evaluation methods (e.g. Fitted Q-Evaluation, Le et al. (2019)).* As for safe optimization, Berkenkamp et al. (2017) propose to ensure safety by keeping the agent within safe regions, which are characterized by a Lyapunov function. However, they assume the environment to be deterministic, i.e., $p(s'|s,a) = 1$ for the successive state $s'$, which is a significant limitation as most MDPs are stochastic. Their method is also model-based, requiring knowledge of the transition functions. *In contrast, our approach copes well with stochastic environments and is model-free.*

Safe reinforcement learning is often modeled as a Constrained Markov Decision Process (CMDP) (Gu et al., 2022; Wachi et al., 2024; Liu et al., 2021), in which we need to maximize the agent reward while making agents satisfy safety constraints. Achiam et al. (2017) enforce a constant threshold to constrain the expected total cost. However, even though they adopt the trust-region method to control policy updates, the expected total cost of the new policy can still exceed the safety threshold at each update step, leading to uncontrolled violations of the safety constraints over time. *In contrast, our method inherently integrates safety constraints into the policy design, ensuring strict satisfaction of constraints throughout execution.* Wachi and Sui (2020) propose a method for safe reinforcement learning in constrained Markov decision processes (CMDPs) by using a Gaussian Process to model the safety constraints and guide exploration. Nevertheless, their approach needs to compute the covariance matrix between explored states throughout the execution, which is computationally expensive, especially in environments with large state spaces. In addition, they assume that the state transitions are deterministic, making their method highly restricted. *In contrast, our algorithm does not rely on knowledge about the complicated covariances and copes with stochastic environments.*

**Variance Reduction.** Variance reduction for policy evaluation in reinforcement learning (RL) is also widely explored. Since using the target policy as the data-collecting policy (i.e., the behavior policy) for evaluating itself is not optimal (Owen, 2013), some recent studies focus on searching for or designing a data-reducing behavior policy without considering safety. Hanna et al. (2017) formulate the task of searching for a variance-reduction behavior policy as an optimization problem. They

parameterize the behavior policy and use stochastic gradient descent to update the policy. However, in reinforcement learning, the stochastic method has been known to easily get stuck in highly suboptimal points in just moderately complex environments (Williams, 1992), where various local optimal and saddle points exist. Besides, they do not consider the safety of their obtained behavior policy, which might cause damage during execution. *In contrast, our method directly learns the globally optimal behavior policy with safety guarantees.* Moreover, to learn the behavior policy effectively, their method requires highly sensitive hyperparameter tuning, where the learning rate can vary by up to $10^5$ times across different environments, as reported by Hanna et al. (2017). This sensitivity requires online tuning, which consumes online data. *In contrast, we introduce an efficient algorithm to learn our behavior policy with purely offline data.* Furthermore, their methods need the online data to be complete trajectories. *In contrast, our algorithm copes well with incomplete offline tuples, making it widely applicable.*

Zhong et al. (2022) also design a variance-reducing behavior policy for policy evaluation. They adjust the behavior policy to focus on under-sampled data segments. However, their method requires the offline data to be in complete trajectories, and the data must be generated by known behavior policies that are highly similar to the target policy. These assumptions are strong. *In contrast, our method effectively handles incomplete offline segments from potentially unknown and diverse behavior policies.* Zhong et al. (2022) also ignore safety constraints in their behavior policy, leading to potential risks in executing the data-collecting behavior policy. *In contrast, our method incorporates safety constraints into the search for the optimal variance-reducing behavior policy, ensuring safety throughout the execution.*

Using the per-decision importance sampling estimator, Mukherjee et al. (2022) also propose a behavior policy to reduce variance. Nevertheless, they also do not consider the crucial safety problem. Besides, their results are restricted to tree-structured MDPs, which is a significant limitation. *In contrast, our method is applicable on general CMDPs, extensions of the widely studied MDPs.* Moreover, Mukherjee et al. (2022) also leverage a model-based approach. The current best method in behavior policy design is proposed by Liu and Zhang (2024), where they find an optimal and an offline-learnable behavior policy. However, all these approaches (Hanna et al., 2017; Zhong et al., 2022; Liu and Zhang, 2024) focus solely on reducing evaluation variance while neglecting a critical issue: safety. Without integrating safety constraints into the design of the behavior policy, its online execution could cause unforeseen and severe damage. *In contrast, we consider the variance minimization problem with safety constraints, obtaining a behavior policy that is safe throughout the execution, while simultaneously achieving substantial variance reduction compared with the on-policy method.*

## 3 BACKGROUND

A finite Markov decision process (MDP, Puterman (2014)) includes a finite state space $\mathcal{S}$, a finite action space $\mathcal{A}$, a reward function $r : \mathcal{S} \times \mathcal{A} \to \mathbb{R}$, a transition probability function $p : \mathcal{S} \times \mathcal{S} \times \mathcal{A} \to [0,1]$, an initial state distribution $p_0 : \mathcal{S} \to [0,1]$, and a constant horizon length $T$. In this paper, to impose safety constraints, we consider constrained Markov decision processes (CMDPs, Altman (2021)), which extends the MDPs with a cost function $c : \mathcal{S} \times \mathcal{A} \to [0, \infty)$. Without loss of generality, we consider the undiscounted setting for simplicity. Our method is compatible with the discounted setting (Puterman, 2014) as long as the horizon is fixed and finite. For any integer, we define a shorthand $[n] \doteq \{0, 1, \ldots, n\}$. For any set $\mathcal{X}$, we use $|\mathcal{X}|$ to denote its cardinality. We use $\Delta^{|\mathcal{X}|-1}$ to denote the $(|\mathcal{X}| - 1)$-dimensional probability simplex, representing the set of all probability distributions over the set $\mathcal{X}$.

A constrained Markov decision process (CMDP) begins at time step 0, where an initial state $S_0$ is sampled from $p_0$. At each time step $t \in [T-1]$, an action $A_t$ is sampled based on $\pi_t(\cdot \mid S_t)$. Here, $\pi_t : \mathcal{A} \times \mathcal{S} \to [0,1]$ denotes the policy at time step $t$. Thereafter, a reward $R_{t+1} \doteq r(S_t, A_t)$ and a cost $C_{t+1} \doteq c(S_t, A_t)$ is emitted by the environment. A successor state $S_{t+1}$ is then sampled from $p(\cdot \mid S_t, A_t)$. We define the abbreviation $\pi_{i:j} \doteq \{\pi_i, \pi_{i+1}, \ldots, \pi_j\}$ and $\pi \doteq \pi_{0:T-1}$. At each time step $t$, the return for the reward $r$ is defined as $G_t \doteq \sum_{i=t+1}^{T} R_i$, and the return for the cost $c$ is $G_t^c \doteq \sum_{i=t+1}^{T} C_i$. Then, we define the state-value and action-value functions for the reward $r$ as $v_{\pi,t}(s) \doteq \mathbb{E}_\pi [G_t \mid S_t = s]$ and $q_{\pi,t}(s,a) \doteq \mathbb{E}_\pi [G_t \mid S_t = s, A_t = a]$. Similarly, the state-value and action-value functions for the cost $c$ are defined as $v_{\pi,t}^c(s) \doteq \mathbb{E}_\pi [G_t^c \mid S_t = s]$ and $q_{\pi,t}^c(s,a) \doteq$

$\mathbb{E}_\pi\left[G_t^c \mid S_t = s, A_t = a\right]$. We adopt the total rewards performance metric from Puterman (2014) to measure the performance of the policy $\pi$, which is defined as $J(\pi) \doteq \sum_s p_0(s)v_{\pi,0}(s)$. Likewise, we also define the total costs of $\pi$ as $J^c(\pi) \doteq \sum_s p_0(s)v_{\pi,0}^c(s)$. In this paper, we use Monte Carlo methods, as introduced by Kakutani (1945), for estimating the total rewards $J(\pi)$. The most straightforward and prevalent technique among many of its variants is to draw samples of $J(\pi)$ through the online execution of the policy $\pi$. As the number of samples increases, the empirical average of the sampled returns is guaranteed to converge to $J(\pi)$. This method is called on-policy learning (Sutton 1988) as it estimates a policy $\pi$ by executing itself,

In this work, we focus on off-policy evaluation, in which the goal is to estimate the total rewards $J(\pi)$ of an interested policy $\pi$, called the *target policy* by executing a different policy $\mu$, called the *behavior policy*. We generate each trajectory $\{S_0, A_0, R_1, C_1, S_1, A_1, R_2, C_2, \ldots, S_{T-1}, A_{T-1}, R_T, C_T\}$ by a behavior policy $\mu$ with $S_0 \sim p_0$, $A_t \sim \mu_t(\cdot|S_t)$. For simplicity, we use a shorthand $\tau_{t:T-1}^{\mu_{t:T-1}}$ for a trajectory generated by the behavior policy $\mu$ from the time step $t$ to the time step $T-1$ inclusively. It is defined as $\tau_{t:T-1}^{\mu_{t:T-1}} \doteq \{S_t, A_t, R_{t+1}, C_{t+1} \ldots, S_{T-1}, A_{T-1}, R_T, C_T\}$. In off-policy evaluation, to give an estimate of $J(\pi)$, we adopt the importance sampling ratio to reweigh rewards collected by the behavior policy $\mu$. We define the importance sampling ratio at time $t$ as $\rho_t \doteq \frac{\pi_t(A_t|S_t)}{\mu_t(A_t|S_t)}$. We also define the product of importance sampling ratios from time $t$ to $t' \geq t$ as $\rho_{t:t'} \doteq \prod_{k=t}^{t'} \frac{\pi_k(A_k|S_k)}{\mu_k(A_k|S_k)}$. Various methods utilize importance sampling ratios within off-policy learning frameworks (Geweke, 1988; Hesterberg, 1995; Koller and Friedman, 2009; Thomas, 2015). In this paper, We study the per-decision importance sampling estimator (PDIS, Precup et al. (2000)). The PDIS Monte Carlo estimator is defined as $G^{\mathrm{PDIS}}(\tau_{t:T-1}^{\mu_{t:T-1}}) \doteq \sum_{k=t}^{T-1} \rho_{t:k}R_{k+1}$. We also use the recursive expression of the PDIS estimator as

$$G^{\mathrm{PDIS}}(\tau_{t:T-1}^{\mu_{t:T-1}}) = \begin{cases} \rho_t\left(R_{t+1} + G^{\mathrm{PDIS}}(\tau_{t+1:T-1}^{\mu_{t+1:T-1}})\right) & t \in [T-2], \\ \rho_t R_{t+1} & t = T-1. \end{cases} \quad (1)$$

With the classic policy coverage assumption (Precup et al., 2000; Maei, 2011; Sutton et al., 2016; Zhang, 2022; Liu et al., 2024b) $\forall t, s, a, \quad \mu_t(a|s) = 0 \implies \pi_t(a|s) = 0$, $G^{\mathrm{PDIS}}$ provides an *unbiased* estimation for $J(\pi)$, i.e., $\mathbb{E}\left[G^{\mathrm{PDIS}}(\tau_{0:T-1}^{\mu_{0:T-1}})\right] = J(\pi)$. Since the PDIS estimator is unbiased, reducing its variance is sufficient for improving its sample efficiency. We achieve this variance reduction by designing and learning proper behavior policies.

## 4 CONSTRAINED VARIANCE MINIMIZATION FOR CONTEXTUAL BANDITS

In this section, we focus on variance minimization in policy evaluation under safety constraints in contextual bandits. These discussions provide the foundation for the more complicated optimization problems in sequential reinforcement learning settings, which we explore in Section 5. Notations defined in this section are independent of the rest of the paper.

We consider contextual bandits as one-step CMDPs, where the trajectories are in the form of $(s, a, r, c)$. To estimate the performance of the target policy $\pi$, $\mathbb{E}_{a\sim\pi}[r(s,a)]$, with data collected by a behavior policy $\mu$, we adopt the importance sampling ratio (Rubinstein, 1981) to reweigh the reward collected by $\mu$. That is, we use $\mathbb{E}_{a\sim\mu}[\rho(a|s)r(s,a)]$ as an estimator, where $\rho(a|s) = \frac{\pi(a|s)}{\mu(a|s)}$. Recall $\Delta^{|\mathcal{A}|-1}$ is the probability simplex representing all probability distributions over the set $\mathcal{A}$. To ensure that this off-policy evaluation is unbiased, a classic choice by Rubinstein (1981) searches for $\mu$ in

$$\Lambda_- \doteq \left\{\mu \mid \forall s, a, \mu(a|s) = 0 \Rightarrow \pi(a|s) = 0 \land \forall s, \mu(\cdot|s) \in \Delta^{|\mathcal{A}|-1}\right\}.$$

In this work, we search in an enlarged space $\Lambda$ (Owen, 2013; Liu and Zhang, 2024), where

$$\Lambda \doteq \left\{\mu \mid \forall s, a, \mu(a|s) = 0 \Rightarrow \pi(a|s)r(s,a) = 0 \land \forall s, \mu(\cdot|s) \in \Delta^{|\mathcal{A}|-1}\right\}. \quad (2)$$

Although a behavior policy $\mu$ in $\Lambda$ may not cover the target policy $\pi$, $\mu$ still gives unbiased estimation in statistics. In the following lemma, we show that searching for $\mu$ in this enlarged space $\Lambda$ guarantees unbiasedness in the contextual bandits setting.

**Lemma 1.** $\forall \mu \in \Lambda$, $\forall s$,

$$\mathbb{E}_{a\sim\mu}[\rho(a|s)r(s,a)] = \mathbb{E}_{a\sim\pi}[r(s,a)].$$

Its proof is in Appendix A.1. Our goal is to search for a variance-minimizing behavior policy $\mu$. Except for the unbiasedness guaranteed by the search space $\Lambda$, we also require $\mu$ to satisfy safety constraints which will be defined later. We formulate the variance minimization objective as, $\forall s$,

$$\min_{\mu \in \Lambda} \quad \mathbb{V}_{a \sim \mu}(\rho(a|s)r(s,a)). \tag{3}$$

Then, with the unbiasedness in Lemma 1, we can further decompose the objective in (3) as

$$\mathbb{V}_{a \sim \mu}(\rho(a|s)r(s,a)) = \mathbb{E}_{a \sim \mu}[(\rho(a|s)r(s,a))^2] - \mathbb{E}_{a \sim \mu}[\rho(a|s)r(s,a)]^2 \tag{4}$$

$$= \mathbb{E}_{a \sim \mu}[\rho(a|s)^2 r(s,a)^2] - \mathbb{E}_{a \sim \pi}[r(s,a)]^2. \quad \text{(By Lemma 1)}$$

Since the second term is a constant and is unrelated to $\mu$, it suffices to solve

$$\min_{\mu \in \Lambda} \quad \mathbb{E}_{a \sim \mu}[\rho(a|s)^2 r(s,a)^2]. \tag{5}$$

Next, to ensure the safety of executing the behavior policy $\mu$, we incorporate a safety constraint into the variance minimization problem. Since measuring safety by the expected cost is a common approach in the safety RL community (Berkenkamp et al., 2017; Achiam et al., 2017; Chow et al., 2018), we require that the expected cost of $\mu$ remains within a threshold related to the expected cost of $\pi$. Given a safety parameter $\epsilon \in [0, \infty)$, define a cost threshold

$$\delta_\epsilon(s) \doteq (1+\epsilon)\mathbb{E}_{a \sim \pi}[c(s,a)].$$

We impose the following constraint to the optimization problem (5)

$$\mathbb{E}_{a \sim \mu}[c(s,a)] \leq \delta_\epsilon(s), \quad \forall s. \tag{6}$$

This constraint requires that the expected cost of the designed behavior policy $\mu$ should be smaller than the multiple of the expected cost of the target policy $\pi$. By satisfying this constraint, we maintain a desired level of safety during the execution of the behavior policy $\mu$. This safety is defined with respect to the target policy $\pi$, which is executed in the classic on-policy evaluation method. By setting $\epsilon = 0$, behavior policies satisfying this constraint are guaranteed to be safer than the target policy.

Notably, another line of research focused on policy safety chooses a constant threshold for the expected cost. We can simply modify (6) into a constant-threshold constraint by replacing the threshold function $\delta_\epsilon(s)$ with a constant $\delta$. However, such absolute thresholds may make optimization problems infeasible. Strong assumptions on environments and policies have to be made to guarantee the existence of feasible solutions under absolute threshold (Achiam et al., 2017). In this paper, we impose the safety constraint with respect to the target policy $\pi$, because our goal is to design a safe behavior policy to address the high variance associated with classic on-policy evaluation methods. The parameter $\epsilon$ in our threshold allows RL practitioners to adjust safety tolerance based on the specific requirements of the problem, as safety constraints are often highly problem-dependent (Achiam et al., 2017). In Section 7, we demonstrate our method in sequential reinforcement learning with a harsh threshold, $\epsilon = 0$, achieving both variance and cost reduction compared to the on-policy method.

We formally define our optimization problem and prove its convexity and feasibility in the following theorem.

**Lemma 2.** *For all $\epsilon$ and $s$, the following optimization problem is convex and feasible.*

$$\min_{\mu \in \Lambda} \quad \mathbb{E}_{a \sim \mu}[\rho(a|s)^2 r(s,a)^2], \tag{7}$$

$$s.t. \quad \mathbb{E}_{a \sim \mu}[c(s,a)] \leq \delta_\epsilon(s). \tag{8}$$

Its proof is in Appendix A.2. Use $\mu^*$ to denote the optimal solution of the above optimization problem. We have the following lemma.

**Lemma 3.** *For all $\epsilon$ and $s$, let $\mu^*$ be the optimal solution of optimization problem (7), we have*

$$\mathbb{V}_{a \sim \mu^*}(\rho(a|s)r(s,a)) \leq \mathbb{V}_{a \sim \pi}(r(s,a)).$$

*Proof.* We first show that the target policy $\pi$ is always in the feasible set of the optimization problem (7). We define the set of feasible policies as

$$\mathcal{F} \doteq \{\mu \in \Lambda \mid \forall \epsilon, s, \mathbb{E}_{a \sim \mu}[c(s,a)] \leq \delta_\epsilon(s)\}. \tag{9}$$

Because $\epsilon \in [0, \infty)$, for the safety constraint, we have

$$\mathbb{E}_{a \sim \pi}[c(s, a)] \leq (1 + \epsilon)\mathbb{E}_{a \sim \pi}[c(s, a)] = \delta_{\epsilon}(s).$$

By the definition of $\Lambda$ (2), $\pi \in \Lambda$. Thus, $\pi \in \mathcal{F}$. Because

$$\mu^* \doteq \underset{\mu \in \mathcal{F}}{\operatorname{argmin}} \, \mathbb{E}_{a \sim \mu}[\rho(a|s)^2 r(s, a)^2] \tag{10}$$

is the optimal solution, we have

$$\begin{aligned}
& \mathbb{V}_{a \sim \mu^*}(\rho(a|s)r(s, a)) \\
=& \mathbb{E}_{a \sim \mu^*}[\rho(a|s)^2 r(s, a)^2] - \mathbb{E}_{a \sim \pi}[r(s, a)]^2 && \text{(by (4))} \\
\leq& \mathbb{E}_{a \sim \pi}[\rho(a|s)^2 r(s, a)^2] - \mathbb{E}_{a \sim \pi}[r(s, a)]^2 && \text{(by (10))} \\
=& \mathbb{E}_{a \sim \pi}[r(s, a)^2] - \mathbb{E}_{a \sim \pi}[r(s, a)]^2 \\
=& \mathbb{V}_{a \sim \pi}(r(s, a)).
\end{aligned}$$

$\square$

In Section 5, we expand Lemma 2 and Lemma 3 from contextual bandits to sequential reinforcement learning in Theorem 1 and Theorem 2. We show that with a recursive expression of the estimation variance, we can reduce the sequential problem into bandits in each time step $t$, and thereafter obtain the optimal behavior policy $\mu^*$ that minimizes variance under safety constraints.

## 5    CONSTRAINED VARIANCE MINIMIZATION FOR SEQUENTIAL REINFORCEMENT LEARNING

In this section, we extend the techniques from contextual bandits to the sequential reinforcement learning setting. We seek to find an optimal behavior policy $\mu$ that reduces the variance $\mathbb{V}\left(G^{\text{PDIS}}(\tau_{0:T-1}^{\mu_{0:T-1}})\right)$ under safety constraints. Before defining the optimization problem, we first define the policy space we search for the behavior policy to ensure the unbiasedness of the PDIS estimator. Conventional methods search $\mu$ in the set of all policies that cover the target policy $\pi$ (Sutton and Barto, 2018), i.e.,

$$\Lambda_- \doteq \{\mu \mid \forall t, s, a, \mu_t(a|s) = 0 \Rightarrow \pi_t(a|s) = 0 \wedge \forall t, s, \mu_t(\cdot|s) \in \Delta^{|\mathcal{A}|-1}\}.$$

In this paper, similar to the bandits setting (2), we search in an enlarged set $\Lambda$, which is defined as

$$\Lambda \doteq \{\mu \mid \forall t, s, a, \mu_t(a|s) = 0 \Rightarrow \pi_t(a|s)q_{\pi,t}(s, a) = 0 \wedge \forall t, s, \mu_t(\cdot|s) \in \Delta^{|\mathcal{A}|-1}\}. \tag{11}$$

The following lemma from Liu and Zhang (2024) ensures the unbiasedness of the off-policy estimator with the behavior policy $\mu \in \Lambda$.

**Lemma 4.** $\forall \mu \in \Lambda$, $\forall t$, $\forall s$,

$$\mathbb{E}\left[G^{PDIS}(\tau_{t:T-1}^{\mu_{t:T-1}}) \mid S_t = s\right] = v_{\pi,t}(s).$$

Its proof is in Appendix A.3. A natural idea to do variance minimization under safety constraints with a safety parameter $\epsilon \in [0, \infty)$ is to solve the following optimization problem

$$\begin{aligned}
\min_{\mu \in \Lambda} \quad & \mathbb{V}\left(G^{\text{PDIS}}(\tau_{0:T-1}^{\mu_{0:T-1}})\right), \tag{12} \\
\text{s.t.} \quad & J^c(\mu) \leq (1 + \epsilon)J^c(\pi),
\end{aligned}$$

where $J^c(\mu) \doteq \sum_s p_0(s)v_{\mu,0}^c(s)$ is the expected cost of the behavior policy $\mu$. Solving this problem directly is very challenging. When designing a policy at a time step $t$, we need to consider not only the immediate reward generated by this action but also the future consequences. Hanna et al. (2017) try to solve this problem without safety constraints by directly optimizing the behavior policy $\mu$ with gradient descent. However, this approach requires online data to optimize $\mu$ and struggles in even moderately complicated environments as shown in Zhong et al. (2022) and Liu and Zhang (2024).

In this paper, we therefore propose to solve this problem in a backward way while ensuring safety constraints. Given an $\epsilon$, use

$$\delta_{\epsilon,t}(s) \doteq (1 + \epsilon)v_{\pi,t}^c(s) \tag{13}$$

to denote the safety threshold. We define an extended reward function $\tilde{r}_t(s,a)$ and a behavior policy $\mu^*$. They are defined in the order of $\left\{ \tilde{r}_{T-1}, \mu^*_{T-1}, \tilde{r}_{T-2}, \mu^*_{T-2}, \cdots, \tilde{r}_0, \mu^*_0 \right\}$. Denote the variance of the state value for the next state given the current state-action pair $(s,a)$ as $\nu_{\pi,t}(s,a)$. We have

$$\nu_{\pi,t}(s,a) \doteq \begin{cases} 0 & t = T-1, \\ \mathbb{V}_{S_{t+1}}\left( v_{\pi,t+1}(S_{t+1}) \mid S_t = s, A_t = a \right) & t \in [T-2]. \end{cases}$$

Then, the extended reward function is defined as

$$\tilde{r}_t(s,a) \doteq \begin{cases} r_{\pi,t}(s,a)^2 & t = T-1, \\ \nu_{\pi,t}(s,a) + q_{\pi,t}(s,a)^2 + \mathbb{E}_{S_{t+1}}\left[ \mathbb{V}\left( G^{\text{PDIS}}(\tau^{\mu^*_{t+1:T-1}}_{t+1:T-1}) \mid S_{t+1} \right) \mid s, a \right] & t \in [T-2]. \end{cases}$$

$$(14)$$

The behavior policy $\mu^*_t$ is defined as the optimal solution to the following problem. $\forall t, s$,

$$\min_{\mu_t \in \Lambda} \quad \mathbb{E}_{a \sim \mu_t}[\rho_t^2 \tilde{r}_t(s,a)],$$
$$\text{s.t.} \quad \mathbb{E}_{a \sim \mu_t}[q^c_{\mu,t}(s,a)] \leq \delta_{\epsilon,t}(s).$$

We have the following theorem showing the convexity and feasibility of (15), thus ensuring the existence of the behavior policy $\mu^*$.

**Theorem 1.** $\forall \epsilon \geq 0$, $\forall t$, $\forall s$, the following optimization problem is convex and feasible.

$$\min_{\mu_t \in \Lambda} \quad \mathbb{E}_{a \sim \mu_t}[\rho_t^2 \tilde{r}_t(s,a)], \tag{15}$$
$$\text{s.t.} \quad \mathbb{E}_{a \sim \mu_t}[q^c_{\mu,t}(s,a)] \leq \delta_{\epsilon,t}(s). \tag{16}$$

Its proof is in Appendix A.4. We notice that the constrained optimization problem (15) is similar to (7), which is the optimization problem introduced in Section 4. In the contextual bandit setting (7), we optimize the objective with respect to the reward function $r$, ensuring variance reduction (Lemma 3). In sequential reinforcement learning (15), we optimize with respect to the extended reward function $\tilde{r}$, achieving variance reduction (Theorem 2 and (17)), while simultaneously guaranteeing safety (18). This observation provides a key insight: the step-wise optimization problem in *sequential reinforcement learning* can be viewed as a reduced optimization problem in one-step *contextual bandits*, where the reward is $\tilde{r}$. In Section 6, we further propose an efficient algorithm to learn $\tilde{r}$ without directly addressing the complicated trajectory variance $\mathbb{V}\left( G^{\text{PDIS}}(\tau^{\mu_{t+1:T-1}}_{t+1:T-1}) \mid S_{t+1} \right)$, making long-horizon RL problems more tractable.

**Theorem 2.** *The behavior policy $\mu^*$ reduces variance compared with the on-policy evaluation method.*

$$\forall t, s, \mathbb{V}\left( G^{\text{PDIS}}(\tau^{\mu^*_{t:T-1}}_{t:T-1}) \mid S_t = s \right) \leq \mathbb{V}\left( G^{\text{PDIS}}(\tau^{\pi_{t:T-1}}_{t:T-1}) \mid S_t = s \right).$$

Its proof is in Appendix A.5. We also present the following theorem to demonstrate variance reduction and safety guarantee with respect to the original constrained optimization problem (12).

**Theorem 3.** *For all $\epsilon \geq 0$, the corresponding behavior policy $\mu^*$ has the following property*

1. $\mathbb{V}\left( G^{\text{PDIS}}(\tau^{\mu^*_{0:T-1}}_{0:T-1}) \right) \leq \mathbb{V}\left( G^{\text{PDIS}}(\tau^{\pi_{0:T-1}}_{0:T-1}) \right)$  (17)

2. $J^c(\mu^*) \leq (1+\epsilon)J^c(\pi)$  (18)

Its proof is in Appendix A.6. Notably, (18) shows that our step-wise safety-constraint (16) is stricter than the original constraint (12).

## 6 Learning the Optimal Behavior Policy

In this section, we propose an efficient algorithm to learn $\tilde{r}$ with previously logged offline data, and subsequently derive the optimal behavior policy $\mu^*$ under safety constraints. We notice that learning $\tilde{r}$ by (14) is inefficient since we need to approximate the complicated variance $\mathbb{V}\left( G^{\text{PDIS}}(\tau^{\mu_{t+1:T-1}}_{t+1:T-1}) \mid S_t \right)$, which involves the entire future trajectory. To tackle this challenge, we present a recursive expression of $\tilde{r}$ in the following lemma.

---

**Algorithm 1:** Safety-Constrained Optimal Policy Evaluation (SCOPE)

---

1: **Input:** a target policy $\pi$,
         an offline dataset $\mathcal{D} = \{(t_i, s_i, a_i, r_i, c_i, s'_i)\}_{i=1}^m$
2: **Output:** a behavior policy $\mu^*$
3: Approximate $q_{\pi,t}, q_{\pi,t}^c$ from $\mathcal{D}$
4: **for** $t = T - 1$ to $0$ **do**
5:      Approximate $\tilde{r}_t$ from $\mathcal{D}$ by Lemma 5
6:      Approximate $\mu_t^*(a|s)$ following (15)
7: **end for**
8: **Return:** the approximated behavior policy $\mu^*$

---

**Lemma 5.** $\forall s, a$, when $t = T - 1$, $\tilde{r}_t(s, a) = r_{\pi,t}(s, a)^2$. When $t \in [T - 2]$,

$$\tilde{r}_t(s, a) = 2q_{\pi,t}(s, a)r(s, a) - r(s, a)^2 + \mathbb{E}_{s' \sim p, a' \sim \pi}\left[\frac{\pi_{t+1}(a'|s')}{\mu_{t+1}^*(a'|s')}\tilde{r}_{\pi,t+1}(s', a')\right].$$

Its proof is in Appendix A.7. With this lemma, we can learn $\tilde{r}$ recursively without approximating the complicated trajectory variance. Then, by (33) in the appendix, we can also decompose the widely interested variance target in a succinct form

$$\underbrace{\mathbb{V}\left(G^{\text{PDIS}}(\tau_{t:T-1}^{\mu_{t:T-1}^*}) \mid S_t = s\right)}_{(a)} = \underbrace{\mathbb{E}_{a \sim \mu}[\rho_t^2 \tilde{r}_t(s, a)]}_{(b)} - \underbrace{v_{\pi,t}(s)^2}_{(c)}, \quad \forall s, t. \tag{19}$$

This succinct form offers a way to approximate the complicated trajectory variance term (a) from (b) and (c), which do not contain any variance term themselves. This is a surprising result because previously the best simplification of the variance for off-policy estimator (a) still depends on state-value variance terms (Jiang and Li, 2016; Liu and Zhang, 2024). With (19), we can approximate the variance of the off-policy estimator in a model-free way with only segmented offline data.

For broad applicability, we adopt the behavior policy-agnostic offline learning setting (Nachum et al., 2019), where the offline data has $m$ previously logged data tuples in the form of $\{(t_i, s_i, a_i, r_i, c_i, s'_i)\}_{i=1}^m$. These data tuples can be generated by one or more possibly unknown behavior policies, and they are not required to form a complete trajectory. In the $i$-th data tuple, $t_i$ is the time step, $s_i$ is the state at time step $t_i$, $a_i$ is the action taken, $r_i$ is the observed reward, $c_i$ is the observed cost, and $s'_i$ is the successor state. In this paper, we learn $\tilde{r}$ from previously logged offline data. Previously logged offline data are cheap and readily available compared with online data. This makes them a great engine for improving policy evaluation in the online phase. Compared with gradient-based methods (Hanna et al., 2017; Zhong et al., 2022) which need complete online trajectories, our method does not require a long online warm-up time to find a good behavior policy because we are able to utilize offline data. Subsequently, the optimal variance-reducing behavior policy $\mu^*$ under safety constraints is approximated through standard convex optimization solvers (Nocedal and Wright, 1999; Agrawal et al., 2018).

## 7 EMPIRICAL RESULTS

In this section, we demonstrate the empirical results comparing our methods against three baselines: **(1)** the on-policy Monte Carlo estimator, **(2)** the robust on-policy sampling estimator (ROS, Zhong et al. (2022)), and **(3)** the offline data informed estimator (ODI, Liu and Zhang (2024)). To ensure our method attains lower cost and is thus even safer than the on-policy estimator, we choose $\epsilon = 0$ in the threshold $\delta_{\epsilon,t}$ (13). All methods learn their required parameters from the same offline dataset to ensure fair comparisons. Given previously logged offline data, our method learns the optimal behavior policy under safety constraints using Algorithm 1.

We name our algorithm Safety-Constrained Optimal Policy Evaluation (SCOPE) to emphasize that safety constraints are inherently considered in the design of the variance-minimizing behavior policy, unlike previous methods that overlook safety concerns. A metaphor for SCOPE is that it builds a bridge focused on efficient transportation (evaluation efficiency) while simultaneously ensuring traffic safety (satisfying safety constraints).

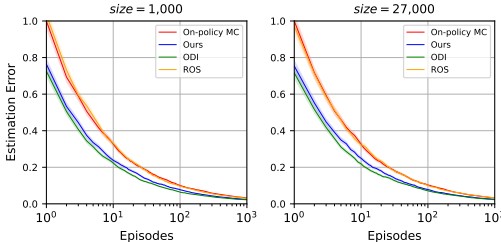 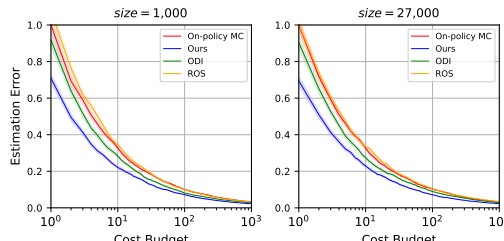

Figure 1: Results on Gridworld with *episodes* as x-axis. Each curve is averaged over 900 runs (30 target policies, each having 30 independent runs). Shaded regions denote standard errors and are invisible for some curves as they are too small.

Figure 2: Results on Gridworld with *cost budget* as x-axis. *Cost budget* is the total cost of execution. Each curve is averaged over 900 runs (30 target policies, each having 30 independent runs). Shaded regions denote standard errors.

| Env Size | On-policy MC | Ours | ODI | ROS |
|---|---|---|---|---|
| 1,000 | 1.000 | **0.861** | 1.602 | 1.083 |
| 27,000 | 1.000 | **0.849** | 1.590 | 1.067 |

Table 1: Average trajectory cost on Gridworld. Numbers are normalized by the cost of the on-policy estimator. ODI and ROS have much larger costs because they both ignore safety constraints. **Our method is the only method achieving both variance reduction and constraint satisfaction.**

**Gridworld:** We first conduct experiments in Gridworld with $n^3$ states. Each Gridworld is an $n \times n$ grid with the time horizon also being $n$. Gridworld environments offer a great tool to test algorithm scalability, because the number of states scales cubically with $n$. Gridworld in our experiments have $n^3 = 1,000$ and $n^3 = 27,000$ number of states, which are the largest Gridworld environments tested among related works (Zhong et al., 2022; Liu and Zhang, 2024). We test all methods on target policies with various performances. The offline data is generated by many different policies to simulate previously logged offline data. In Figure 1, we report the estimation error against episodes. The estimation error for any line is the absolute error normalized by the absolute error of the on-policy estimator after the first episode. Thus, the estimation error of the on-policy estimator starts from 1. In Figure 2, we report the estimation error against the total cost of execution.

If considering *solely* variance reduction, Figure 1 shows our method outperforms the on-policy estimator and ROS by a large margin. Admittedly, ODI (Liu and Zhang, 2024) is slightly better than our method in terms of variance reduction. However, this slight advantage comes with a huge *trade-off* of safety. As shown in Table 1, ODI has a much larger cost than on-policy evaluation method (more than 1.5 times) and our method (almost twice as much). **This addresses the underestimated fact—solely reducing variance without safety constraints leads to high-cost (unsafe) methods.**

To further demonstrate the superiority of balancing variance reduction and safety cost of our method, we provide Figure 2 to compare the variance reduction each method achieves with the same cost budget. Since our method SCOPE is optimal for safety-constrained variance minimization, it consistently outperforms all baselines in Figure 2, as shown by the lowest blue line. This means that compared with existing best-performing methods, SCOPE needs less cost to achieve the same level of accuracy. From Figure 2, we compute that to achieve the same accuracy that the on-policy estimator achieves with 1000 costs (each on-policy episode has expected cost 1 by normalization), ODI costs $880$ and SCOPE costs only $425$. Following this computation, our method saves $57.5\%$ of costs compared to the on-policy method, and $50\%$ compared to ODI. **This reinforces the underestimated fact from the opposite direction—ensuring safety constraints along with the variance minimization leads to a low-cost method.** Also, notably, our estimator outperforms the on-policy and ROS estimators in *reducing both variance and cost*.

**MuJoCo:** Next, we conduct experiments in MuJoCo robot simulation tasks (Todorov et al., 2012). MuJoCo is a physics engine with a variety of stochastic environments The goal is to control a robot to achieve different behaviors such as walking, jumping, and balancing.

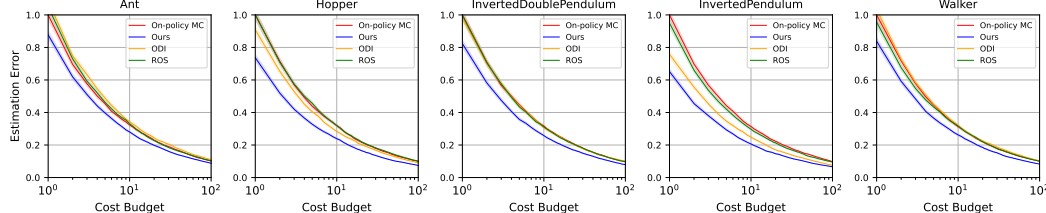

Figure 3: Results on MuJoCo. *Cost budget* on the x-axis is the total cost of execution. Each curve is averaged over 900 runs (30 of target policies, each having 30 independent runs). Shaded regions denote standard errors and are invisible for some curves because they are too small. Results with a larger x-axis range are in the appendix.

|  | On-policy MC | **Ours** | ODI | ROS | Saved Cost Percentage |
|---|---|---|---|---|---|
| Ant | 1000 | **746** | 1136 | 1063 | (1000 - 746)/1000 = **25.4%** |
| Hopper | 1000 | **552** | 824 | 1026 | (1000 - 552)/1000 = **44.8%** |
| I. D. Pendulum | 1000 | **681** | 1014 | 1003 | (1000 - 681)/1000 = **31.9%** |
| I. Pendulum | 1000 | **425** | 615 | 890 | (1000 - 425)/1000 = **57.5%** |
| Walker | 1000 | **694** | 1031 | 960 | (1000 - 694)/1000 = **30.6%** |

Table 2: Cost needed to achieve the same estimation accuracy that on-policy Monte Carlo achieves with 1000 episodes on MuJoCo. Each curve is averaged over 900 runs. Standard errors are plotted in Figure 3.

As confirmed in Table 5 and Table 6 in the appendix, our method is the only method consistently achieving both variance reduction and safety constraint satisfaction. Figure 3 again indicates that our method consistently outperforms all baselines on reducing variance under the same cost budget. This advantage is observed across all five environments, demonstrating the stableness of our method in balancing variance reduction and cost management. Numerically, in Table 2, we show that our method, SCOPE, saves up to $57.5\%$ cost to achieve the desired evaluation accuracy. More experiment details are in Appendix B. It is worth mentioning that our method is robust to hyperparameter choices—all hyperparameters in our method are tuned offline and stay the same across all environments.

## 8 CONCLUSION

In reinforcement learning, due to the sequential nature, policy evaluation often suffers from large variance and thus requires massive data to achieve the desired level of accuracy. In addition, safety is a critical concern for policy execution, since unsafe actions can lead to significant risks and irreversible damage. In this paper, we address these two challenges simultaneously: we propose an optimal variance-minimizing behavior policy under safety constraints.

Theoretically, we show that our estimate is unbiased. Moreover, while simultaneously satisfying safety constraints, our behavior policy is proven to achieve lower variance than the classic on-policy evaluation method (Theorem 2, Theorem 3). We solve the constrained optimization problem without approximating the complicated trajectory variance (Lemma 5), pointing out a promising direction for addressing long-horizon sequential reinforcement learning challenges.

Empirically, compared with existing best-performing methods, we show our method is the only one that achieves both substantial variance reduction and constraint satisfaction for policy evaluation in sequential reinforcement learning. Moreover, it is even superior to previous methods in both variance reduction and execution safety.

Admittedly, as there is no free lunch, if the offline data size is too small—perhaps containing merely a single data tuple—the learned behavior policy in our method may be inaccurate. In this case, for a safe backup, we recommend the on-policy evaluation method. The future work of our paper is to extend the constrained variance minimization technique to temporal difference learning.

ACKNOWLEDGEMENTS

This work is supported in part by the US National Science Foundation (NSF) under grants III-2128019 and SLES-2331904.

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

## A    PROOFS

### A.1    PROOF OF LEMMA 1

*Proof.* $\forall s, \forall \mu \in \Lambda$,

$$
\begin{aligned}
\mathbb{E}_{a\sim\mu}[\rho(a|s)r(s,a)] &= \sum_{a\in\{a|\mu(a|s)>0\}} \mu(a|s)\frac{\pi(a|s)}{\mu(a|s)}r(s,a) \\
&= \sum_{a\in\{a|\mu(a|s)>0\}} \pi(a|s)r(s,a) \\
&= \sum_{a\in\{a|\mu(a|s)>0\}} \pi(a|s)r(s,a) + \sum_{a\in\{a|\mu(a|s)=0\}} \pi(a|s)r(s,a) \qquad (\mu\in\Lambda) \\
&= \sum_{a} \pi(a|s)r(s,a) \\
&= \mathbb{E}_{a\sim\pi}[r(s,a)].
\end{aligned}
$$

$\square$

### A.2    PROOF OF LEMMA 2

*Proof.* To prove Lemma 2, we express the objective function as

$$
\mathbb{E}_{a\sim\mu}[\rho(a|s)^2 r(s,a)^2] = \sum_{a\in\{a|\mu(a|s)>0\}} \frac{\pi(a|s)^2 r(s,a)^2}{\mu(a|s)}.
$$

To prove the problem is convex, we begin by examining the feasible set of each constraint separately.

In the first constraint of $\Lambda$ (2),

$$
\forall s, a, \mu(a|s) = 0 \implies \pi(a|s)r(s,a) = 0. \tag{20}
$$

The feasible set of (20) is a linear subspace of $\mathbb{R}^{|\mathcal{A}|}$ defined by a set of linear equations. Thus, this feasible set is convex.

Next, we decompose the other constraint of $\Lambda$ (2), $\mu(\cdot|s) \in \Delta^{|\mathcal{A}|-1}\ \forall s$, into two subconstraints:

$$
\sum_{a} \mu(a|s) = 1, \tag{21}
$$

$$
\forall a, \mu(a|s) \geq 0. \tag{22}
$$

For all $s$, the feasible set in (21) can be written in the vector form as

$$
\mathbf{1}^T \overrightarrow{\mu_s} = 1, \tag{23}
$$

where $\mathbf{1} \in \mathbb{R}^{|\mathcal{A}|}$ is the vector of ones defined as

$$
\mathbf{1} \doteq \begin{bmatrix} 1 \\ \vdots \\ 1 \end{bmatrix},
$$

and $\overrightarrow{\mu_s} \in \mathbb{R}^{|\mathcal{A}|}$ is defined as

$$
\overrightarrow{\mu_s} \doteq \begin{bmatrix} \mu(a_1|s) \\ \vdots \\ \mu(a_{|\mathcal{A}|}|s) \end{bmatrix}.
$$

Since (23) is linear, the constraint (21) is affine and thus convex (Boyd et al., 2004).

For all $s$, the feasible set of (22) is the non-negative orthant, defined as

$$
\mathbb{R}_+^{|\mathcal{A}|} \doteq \{\mu(\cdot|s) \in \mathbb{R}^{|\mathcal{A}|} \mid \mu(a|s) \geq 0, \forall a\}.
$$

Since the non-negative orthant forms a convex cone and is known to be a convex set (Boyd et al., 2004), we conclude that this constraint's feasible set is convex.

Next, we define the vector of costs for all $s$ as

$$\mathbf{c}_s \doteq \begin{bmatrix} c(s, a_1) \\ \vdots \\ c(s, a_{|\mathcal{A}|}) \end{bmatrix}.$$

Then, for all $\epsilon$ and $s$, the safety constraint (8) can be rewritten as

$$\mathbf{c}_s^\top \overrightarrow{\mu_s} \leq \delta_\epsilon(s),$$

which is a linear inequality in $\mu$. Thus, its feasible set is in a convex half-space. Because all the constraints are convex, we conclude that the feasible set $\mathcal{F}$ in (9) is convex.

Finally, we examine the minimization objective (7), where $\pi$ and $r$ are fixed and independent of the behavior policy $\mu$. For all $s$, we express the objective function as

$$\mathbb{E}_{a \sim \mu}[\rho(a|s)^2 r(s, a)^2] = \sum_{a \in \{a | \mu(a|s) > 0\}} \frac{\pi(a|s)^2 r(s, a)^2}{\mu(a|s)}.$$

Then, for each $a$, we decompose the objective function as

$$f_a(\mu(a|s)) \doteq \frac{\pi(a|s)^2 r(s, a)^2}{\mu(a|s)}. \tag{24}$$

Taking the first and second derivatives of $f_a$, we get

$$f_a'(\mu(a|s)) = -\frac{\pi(a|s)^2 r(s, a)^2}{\mu(a|s)^2},$$

$$f_a''(\mu(a|s)) = \frac{2\pi(a|s)^2 r(s, a)^2}{\mu(a|s)^3}.$$

Since $\forall s, a, f_a''(\mu(a|s)) \geq 0$, we know that (24) is convex for all $a$. Then, as a summation of convex functions, (7) is also convex. In conclusion, by the convexity of the feasible set $\mathcal{F}$ and the objective function (7), we obtain the convexity of the constrained optimization problem in Lemma 2.

For feasibility, note that by Lemma 3, $\pi \in \mathcal{F}$, which is the feasible set. Thus, we confirm the feasibility in Lemma 2.

$\square$

### A.3 PROOF OF LEMMA 4

*Proof.* We proceed via induction. For $t = T - 1$, we have

$$\begin{aligned} \mathbb{E}\left[G^{\text{PDIS}}(\tau_{t:T-1}^{\mu_{t:T-1}}) \mid S_t\right] &= \mathbb{E}\left[\rho_t R_{t+1} \mid S_t\right] = \mathbb{E}\left[\rho_t q_{\pi,t}(S_t, A_t) \mid S_t\right] \\ &= \mathbb{E}_{A_t \sim \pi_t(\cdot|S_t)}\left[q_{\pi,t}(S_t, A_t) \mid S_t\right] \qquad \text{(Lemma 1)} \\ &= v_{\pi,t}(S_t). \end{aligned}$$

For $t \in [T-2]$, we have

$$\begin{aligned} &\mathbb{E}\left[G^{\text{PDIS}}(\tau_{t:T-1}^{\mu_{t:T-1}}) \mid S_t\right] \\ =& \mathbb{E}\left[\rho_t R_{t+1} + \rho_t G^{\text{PDIS}}(\tau_{t+1:T-1}^{\mu_{t+1:T-1}}) \mid S_t\right] \\ =& \mathbb{E}\left[\rho_t R_{t+1} \mid S_t\right] + \mathbb{E}\left[\rho_t G^{\text{PDIS}}(\tau_{t+1:T-1}^{\mu_{t+1:T-1}}) \mid S_t\right] \\ =& \mathbb{E}\left[\rho_t R_{t+1} \mid S_t\right] + \mathbb{E}_{A_t \sim \mu_t(\cdot|S_t), S_{t+1} \sim p(\cdot|S_t, A_t)}\left[\mathbb{E}\left[\rho_t G^{\text{PDIS}}(\tau_{t+1:T-1}^{\mu_{t+1:T-1}}) \mid S_t, A_t, S_{t+1}\right] \mid S_t\right] \\ &\hspace{9cm} \text{(Law of total expectation)} \\ =& \mathbb{E}\left[\rho_t R_{t+1} \mid S_t\right] + \mathbb{E}_{A_t \sim \mu_t(\cdot|S_t), S_{t+1} \sim p(\cdot|S_t, A_t)}\left[\rho_t \mathbb{E}\left[G^{\text{PDIS}}(\tau_{t+1:T-1}^{\mu_{t+1:T-1}}) \mid S_{t+1}\right] \mid S_t\right] \\ &\hspace{6cm} \text{(Conditional independence and Markov property)} \end{aligned}$$

$$
\begin{aligned}
=&\mathbb{E}\left[\rho_t R_{t+1} \mid S_t\right] + \mathbb{E}_{A_t \sim \mu_t(\cdot|S_t), S_{t+1} \sim p(\cdot|S_t, A_t)}\left[\rho_t v_{\pi, t+1}(S_{t+1}) \mid S_t\right] && \text{(Inductive hypothesis)} \\
=&\mathbb{E}_{A_t \sim \mu_t(\cdot|S_t)}\left[\rho_t q_{\pi, t}(S_t, A_t) \mid S_t\right] && \text{(Definition of } q_{\pi, t}) \\
=&\mathbb{E}_{A_t \sim \pi_t(\cdot|S_t)}\left[q_{\pi, t}(S_t, A_t) \mid S_t\right] && \text{(Lemma 1)} \\
=&v_{\pi, t}(S_t),
\end{aligned}
$$

which completes the proof. $\qquad\square$

### A.4 PROOF OF THEOREM 1

*Proof.* We first define the set of feasible policies as

$$
\mathcal{F} \doteq \{\mu \in \Lambda \mid \forall \epsilon, t, s, \mathbb{E}_{a \sim \mu_t}[v^c_{\mu, t}(s)] \le \delta_{\epsilon, t}(s)\}. \tag{25}
$$

We begin by examining each constraint. In the first constraint of $\Lambda$ (11),

$$
\forall t, s, a, \mu_t(a|s) = 0 \implies \pi_t(a|s) q_{\pi, t}(s, a) = 0. \tag{26}
$$

The feasible set of (26) is a linear subspace of $\mathbb{R}^{|\mathcal{A}|}$ defined by a set of linear equations. Thus, this feasible set is convex.

Next, we decompose the other constraint of $\Lambda$ (11), $\mu_t(\cdot|s) \in \Delta^{|\mathcal{A}|-1}$, into two constraints:

$$
\sum_a \mu_t(a|s) = 1, \tag{27}
$$

$$
\forall a, \mu_t(a|s) \ge 0. \tag{28}
$$

For all $t$ and $s$, in (27), the feasible set can be written as

$$
\mathbf{1}^\top \overrightarrow{\mu_{s,t}} = 1, \tag{29}
$$

where $\mathbf{1} \in \mathbb{R}^{|\mathcal{A}|}$ is the vector of ones and $\overrightarrow{\mu_{s,t}} \in \mathbb{R}^{|\mathcal{A}|}$ is defined as

$$
\overrightarrow{\mu_{s,t}} \doteq \begin{bmatrix} \mu_t(a_1|s) \\ \vdots \\ \mu_t(a_{|\mathcal{A}|}|s) \end{bmatrix}.
$$

Since (29) is linear, the feasible set of constraint (27) is affine and thus convex (Boyd et al., 2004).

For all $t$ and $s$, the feasible set for the constraint in (28) is the non-negative orthant, defined as

$$
\mathbb{R}^{|\mathcal{A}|}_+ \doteq \{\mu_t(\cdot|s) \in \mathbb{R}^{|\mathcal{A}|} \mid \mu_t(a|s) \ge 0, \forall a\}.
$$

Since the non-negative orthant forms a convex cone and is known to be a convex set (Boyd et al., 2004), we conclude that this constraint is convex.

Next, we define the vector of the state-action value function for the cost $c$ for each $s$ as

$$
\mathbf{q}_{\mu, t} \doteq \begin{bmatrix} q^c_{\mu, t}(s, a_1) \\ \vdots \\ q^c_{\mu, t}(s, a_{|\mathcal{A}|}) \end{bmatrix}.
$$

Then, for all $\epsilon$, $t$ and $s$, the safety constraint (16) can be rewritten as

$$
\mathbf{q}_{\mu, t}^\top \overrightarrow{\mu_{s,t}} \le \delta_{\epsilon, t}(s),
$$

which is a linear inequality in $\mu_t$. Thus, its feasible set is a convex half-space. Because all the constraints' feasible sets are convex, we conclude that the feasible set $\mathcal{F}$ in (25) is convex.

To prove Theorem 1, we express the objective function as

$$
\mathbb{E}_{a \sim \mu_t}[\rho_t^2 \tilde{r}_t(s, a)] = \sum_{a \in \{a | \mu_t(a|s) > 0\}} \frac{\pi_t(a|s)^2 \tilde{r}_t(s, a)}{\mu_t(a|s)},
$$

where $\tilde{r}$ in (14) is defined as

$$\tilde{r}_t(s,a) \doteq \begin{cases} r_{\pi,t}(s,a)^2 & t = T-1, \\ \nu_{\pi,t}(s,a) + q_{\pi,t}(s,a)^2 + \mathbb{E}_{S_{t+1}}\left[\mathbb{V}\left(G^{\text{PDIS}}(\tau_{t+1:T-1}^{\mu_{t+1:T-1}^*}) \mid S_{t+1}\right) \mid s,a\right] & t \in [T-2]. \end{cases}$$

Here, $\tilde{r}_t$ can be learned with logged offline data, as detailed in Algorithm 1, and it is unrelated to $\mu_t$. Then, for each $a$, we decompose the objective function as

$$f_a(\mu_t(a|s)) \doteq \frac{\pi_t(a|s)^2 \tilde{r}_t(s,a)}{\mu_t(a|s)}. \tag{30}$$

Taking the first and second derivatives of $f_a$, we get

$$f_a'(\mu_t(a|s)) = -\frac{\pi_t(a|s)^2 \tilde{r}(s,a)}{\mu_t(a|s)^2},$$

$$f_a''(\mu_t(a|s)) = \frac{2\pi_t(a|s)^2 \tilde{r}(s,a)}{\mu_t(a|s)^3}.$$

Notice that the extended reward $\tilde{r}$ defined in (14) is non-negative, since all the summands are non-negative. Thus, $\forall t, s, a, f_a''(\mu_t(a|s)) \geq 0$, and we know that (30) is convex for all $a$. Then, as a summation of convex functions, (15) is also convex. In conclusion, by the convexity of the feasible set $\mathcal{F}$ and the objective function (15), we obtain the convexity of the constrained optimization problem in Theorem 1.

For feasibility, we show that the set of feasible policies (25) is non-empty. Because $\epsilon \in [0, \infty)$, for the safety constraint, we have

$$\mathbb{E}_{a\sim\pi_t}[v_{\mu,t}^c(s)] \leq (1+\epsilon)\mathbb{E}_{a\sim\pi_t}[v_{\mu,t}^c(s)] = \delta_{\epsilon,t}(s).$$

By the definition of $\Lambda$ (11), $\forall t, \pi_t \in \Lambda$. Therefore, the set of feasible policies (25) is non-empty. Thus, the constrained optimization problem in Theorem 1 is feasible. $\square$

## A.5 PROOF OF THEOREM 2

To prove Theorem 2, we first restate a recursive expression of the variance $\mathbb{V}\left(G^{\text{PDIS}}(\tau_{t:T-1}^{\mu_{t:T-1}}) \mid S_t\right)$ for all $\mu \in \Lambda$ from Liu and Zhang (2024), and present its proof for completeness.

**Lemma 6.** *For any $\mu \in \Lambda$, for $t = T-1$,*

$$\mathbb{V}\left(G^{\text{PDIS}}(\tau_{t:T-1}^{\mu_{t:T-1}}) \mid S_t\right) = \mathbb{E}_{A_t\sim\mu_t}\left[\rho_t^2 q_{\pi,t}^2(S_t, A_t) \mid S_t\right] - v_{\pi,t}^2(S_t),$$

*for $t \in [T-2]$,*

$$\mathbb{V}\left(G^{\text{PDIS}}(\tau_{t:T-1}^{\mu_{t:T-1}}) \mid S_t\right)$$
$$= \mathbb{E}_{A_t\sim\mu_t}\left[\rho_t^2\left(\mathbb{E}_{S_{t+1}}\left[\mathbb{V}\left(G^{\text{PDIS}}(\tau_{t+1:T-1}^{\mu_{t+1:T-1}}) \mid S_t\right) \mid S_t, A_t\right] + \nu_{\pi,t}(S_t, A_t) + q_{\pi,t}^2(S_t, A_t)\right) \mid S_t\right]$$
$$- v_{\pi,t}^2(S_t).$$

*Proof.* For completeness, we provide the proof from Liu and Zhang (2024). We proceed via induction. When $t = T-1$, we have

$$\mathbb{V}\left(G^{\text{PDIS}}(\tau_{t:T-1}^{\mu_{t:T-1}}) \mid S_t\right) = \mathbb{V}\left(\rho_t r(S_t, A_t) \mid S_t\right)$$
$$= \mathbb{V}\left(\rho_t q_{\pi,t}(S_t, A_t) \mid S_t\right)$$
$$= \mathbb{E}_{A_t}\left[\rho_t^2 q_{\pi,t}(S_t, A_t)^2 \mid S_t\right] - v_{\pi,t}(S_t)^2,$$

When $t \in [T-2]$, we have

$$\mathbb{V}\left(G^{\text{PDIS}}(\tau_{t:T-1}^{\mu_{t:T-1}}) \mid S_t\right) \tag{31}$$
$$= \mathbb{E}_{A_t}\left[\mathbb{V}\left(G^{\text{PDIS}}(\tau_{t:T-1}^{\mu_{t:T-1}}) \mid S_t, A_t\right) \mid S_t\right] + \mathbb{V}_{A_t}\left(\mathbb{E}\left[G^{\text{PDIS}}(\tau_{t:T-1}^{\mu_{t:T-1}}) \mid S_t, A_t\right] \mid S_t\right)$$
$$\text{(Law of total variance)}$$

$$=\mathbb{E}_{A_t}\left[\rho_t^2\mathbb{V}\left(r(S_t,A_t)+G^{\text{PDIS}}(\tau_{t+1:T-1}^{\mu_{t+1:T-1}})\mid S_t,A_t\right)\mid S_t\right]$$
$$+\mathbb{V}_{A_t}\left(\rho_t\mathbb{E}\left[r(S_t,A_t)+G^{\text{PDIS}}(\tau_{t+1:T-1}^{\mu_{t+1:T-1}})\mid S_t,A_t\right]\mid S_t\right)\qquad\text{(By (1))}$$
$$=\mathbb{E}_{A_t}\left[\rho_t^2\mathbb{V}\left(G^{\text{PDIS}}(\tau_{t+1:T-1}^{\mu_{t+1:T-1}})\mid S_t,A_t\right)\mid S_t\right]+\mathbb{V}_{A_t}\left(\rho_t\mathbb{E}\left[r(S_t,A_t)+G^{\text{PDIS}}(\tau_{t+1:T-1}^{\mu_{t+1:T-1}})\mid S_t,A_t\right]\mid S_t\right)$$
$$\text{(Deterministic reward }r)$$
$$=\mathbb{E}_{A_t}\left[\rho_t^2\mathbb{V}\left(G^{\text{PDIS}}(\tau_{t+1:T-1}^{\mu_{t+1:T-1}})\mid S_t,A_t\right)\mid S_t\right]+\mathbb{V}_{A_t}\left(\rho_t q_{\pi,t}(S_t,A_t)\mid S_t\right).$$

Further decomposing the first term, we have

$$\mathbb{V}\left(G^{\text{PDIS}}(\tau_{t+1:T-1}^{\mu_{t+1:T-1}})\mid S_t,A_t\right)\qquad\qquad(32)$$
$$=\mathbb{E}_{S_{t+1}}\left[\mathbb{V}\left(G^{\text{PDIS}}(\tau_{t+1:T-1}^{\mu_{t+1:T-1}})\mid S_t,A_t,S_{t+1}\right)\mid S_t,A_t\right]$$
$$+\mathbb{V}_{S_{t+1}}\left(\mathbb{E}\left[G^{\text{PDIS}}(\tau_{t+1:T-1}^{\mu_{t+1:T-1}})\mid S_t,A_t,S_{t+1}\right]\mid S_t,A_t\right)\qquad\text{(Law of total variance)}$$
$$=\mathbb{E}_{S_{t+1}}\left[\mathbb{V}\left(G^{\text{PDIS}}(\tau_{t+1:T-1}^{\mu_{t+1:T-1}})\mid S_{t+1}\right)\mid S_t,A_t\right]+\mathbb{V}_{S_{t+1}}\left(\mathbb{E}\left[G^{\text{PDIS}}(\tau_{t+1:T-1}^{\mu_{t+1:T-1}})\mid S_{t+1}\right]\mid S_t,A_t\right)$$
$$\text{(Markov property)}$$
$$=\mathbb{E}_{S_{t+1}}\left[\mathbb{V}\left(G^{\text{PDIS}}(\tau_{t+1:T-1}^{\mu_{t+1:T-1}})\mid S_{t+1}\right)\mid S_t,A_t\right]+\mathbb{V}_{S_{t+1}}\left(v_{\pi,t+1}(S_{t+1})\mid S_t,A_t\right).\qquad\text{(Lemma 4)}$$

Then, plugging (32) back to (31) yields

$$\mathbb{V}\left(G^{\text{PDIS}}(\tau_{t:T-1}^{\mu_{t:T-1}})\mid S_t\right)$$
$$=\mathbb{E}_{A_t}\left[\rho_t^2\left(\mathbb{E}_{S_{t+1}}\left[\mathbb{V}\left(G^{\text{PDIS}}(\tau_{t+1:T-1}^{\mu_{t+1:T-1}})\mid S_{t+1}\right)\mid S_t,A_t\right]+\mathbb{V}_{S_{t+1}}(v_{\pi,t}(S_{t+1})\mid S_t=s,A_t=a)\right)\mid S_t\right]$$

$$+\mathbb{V}_{A_t}\left(\rho_t q_{\pi,t}(S_t,A_t)\mid S_t\right)$$
$$=\mathbb{E}_{A_t}\left[\rho_t^2\left(\mathbb{E}_{S_{t+1}}\left[\mathbb{V}\left(G^{\text{PDIS}}(\tau_{t+1:T-1}^{\mu_{t+1:T-1}})\mid S_{t+1}\right)\mid S_t,A_t\right]+\mathbb{V}_{S_{t+1}}(v_{\pi,t}(S_{t+1})\mid S_t=s,A_t=a)\right)\mid S_t\right]$$

$$+\mathbb{E}_{A_t}\left[\rho_t^2 q_{\pi,t}(S_t,A_t)^2\mid S_t\right]-\left(\mathbb{E}_{A_t}\left[\rho_t q_{\pi,t}(S_t,A_t)\mid S_t\right]\right)^2$$
$$=\mathbb{E}_{A_t}\left[\rho_t^2\left(\mathbb{E}_{S_{t+1}}\left[\mathbb{V}\left(G^{\text{PDIS}}(\tau_{t+1:T-1}^{\mu_{t+1:T-1}})\mid S_{t+1}\right)\mid S_t,A_t\right]+\mathbb{V}_{S_{t+1}}(v_{\pi,t}(S_{t+1})\mid S_t=s,A_t=a)\right)\mid S_t\right]$$

$$+\mathbb{E}_{A_t}\left[\rho_t^2 q_{\pi,t}(S_t,A_t)^2\mid S_t\right]-v_{\pi,t}(S_t)^2,\qquad\qquad\text{(Lemma 1)}$$
$$=\mathbb{E}_{A_t}\left[\rho_t^2\left(\mathbb{E}_{S_{t+1}}\left[\mathbb{V}\left(G^{\text{PDIS}}(\tau_{t+1:T-1}^{\mu_{t+1:T-1}})\mid S_{t+1}\right)\mid S_t,A_t\right]+\nu_{\pi,t}(S_t,A_t)+q_{\pi,t}(S_t,A_t)^2\right)\mid S_t\right]$$
$$-v_{\pi,t}(S_t)^2,\qquad\qquad\text{(Definition of }\nu)$$

which completes the proof. $\qquad\qquad\qquad\qquad\qquad\qquad\qquad\qquad\qquad\qquad\qquad\square$

Then, with the extended reward $\tilde{r}$ in (14) defined as

$$\tilde{r}_t(s,a)\doteq\begin{cases}r_{\pi,t}(s,a)^2 & t=T-1,\\\nu_{\pi,t}(s,a)+q_{\pi,t}(s,a)^2+\mathbb{E}_{S_{t+1}}\left[\mathbb{V}\left(G^{\text{PDIS}}(\tau_{t+1:T-1}^{\mu_{t+1:T-1}^*})\mid S_{t+1}\right)\mid s,a\right] & t\in[T-2],\end{cases}$$

we can express the variance in a succinct form

$$\mathbb{V}\left(G^{\text{PDIS}}(\tau_{t:T-1}^{\mu_{t:T-1}^*})\mid S_t=s\right)=\mathbb{E}_{a\sim\mu}[\rho_t^2\tilde{r}_t(s,a)]-v_{\pi,t}(s)^2,\quad\forall s,t.\qquad(33)$$

Now, we restate Theorem 2 and present its proof.

**Theorem 2.** *The behavior policy $\mu^*$ reduces variance compared with the on-policy evaluation method.*

$$\forall t,s,\mathbb{V}\left(G^{\text{PDIS}}(\tau_{t:T-1}^{\mu_{t:T-1}^*})\mid S_t=s\right)\leq\mathbb{V}\left(G^{\text{PDIS}}(\tau_{t:T-1}^{\pi_{t:T-1}})\mid S_t=s\right).$$

In Appendix A.4, we show that $\forall t,\pi_t\in\mathcal{F}$, where $\mathcal{F}$ in (25) is the set of feasible policies for the constrained optimization problem in Theorem 1. Recall that $\mu_t^*$ is defined as the optimal solution to the problem (15), i.e.,

$$\mu_t^*\doteq\underset{\mu_t\in\mathcal{F}}{\arg\min}\,\mathbb{E}_{a\sim\mu_t}[\rho_t^2\tilde{r}(s,a)].\qquad\qquad(34)$$

Thus, $\forall t, s$,

$$\mathbb{V}\left(G^{\text{PDIS}}(\tau_{t:T-1}^{\mu_{t:T-1}^*}) \mid S_t = s\right)$$

$$= \mathbb{E}_{a \sim \mu_t^*}[\rho_t^2 \tilde{r}_t(s, a)] - v_{\pi,t}(s)^2 \qquad \text{(By (33))}$$

$$\leq \mathbb{E}_{a \sim \pi_t}[\rho_t^2 \tilde{r}_t(s, a)] - v_{\pi,t}(s)^2 \qquad \text{(By (34) and } \pi_t \in \mathcal{F})$$

$$= \mathbb{V}\left(G^{\text{PDIS}}(\tau_{t:T-1}^{\pi_{t:T-1}}) \mid S_t = s\right), \qquad \text{(By (33))}$$

which completes the proof.

### A.6 PROOF OF THEOREM 3

*Proof.* We first prove the variance reduction property.

$$\mathbb{V}\left(G^{\text{PDIS}}(\tau_{0:T-1}^{\mu_{0:T-1}^*})\right)$$

$$= \mathbb{E}_{S_0}\left[\mathbb{V}\left(G^{\text{PDIS}}(\tau_{0:T-1}^{\mu_{0:T-1}^*}) \mid S_0\right)\right] + \mathbb{V}_{S_0}\left(\mathbb{E}\left[G^{\text{PDIS}}(\tau_{0:T-1}^{\mu_{0:T-1}^*}) \mid S_0\right]\right) \qquad \text{(Law of Total Variance)}$$

$$= \mathbb{E}_{S_0}\left[\mathbb{V}\left(G^{\text{PDIS}}(\tau_{0:T-1}^{\mu_{0:T-1}^*}) \mid S_0\right)\right] + \mathbb{V}_{S_0}\left(v_{\pi,0}(S_0)\right) \qquad \text{(By Lemma 4 and } \mu^* \in \Lambda)$$

$$\leq \mathbb{E}_{S_0}\left[\mathbb{V}\left(G^{\text{PDIS}}(\tau_{0:T-1}^{\pi_{0:T-1}}) \mid S_0\right)\right] + \mathbb{V}_{S_0}\left(v_{\pi,0}(S_0)\right) \qquad \text{(Theorem 2)}$$

$$= \mathbb{E}_{S_0}\left[\mathbb{V}\left(G^{\text{PDIS}}(\tau_{0:T-1}^{\pi_{0:T-1}}) \mid S_0\right)\right] + \mathbb{V}_{S_0}\left(\mathbb{E}\left[G^{\text{PDIS}}(\tau_{0:T-1}^{\pi_{0:T-1}}) \mid S_0\right]\right) \qquad \text{(By Lemma 4 and } \pi \in \Lambda)$$

$$= \mathbb{V}\left(G^{\text{PDIS}}(\tau_{0:T-1}^{\pi_{0:T-1}})\right). \qquad \text{(Law of Total Variance)}$$

Next, we prove the safety constraint satisfaction.

$$J^c(\mu^*)$$

$$= \sum_s p_0(s) v_{\mu^*,0}^c(s)$$

$$= \sum_s p_0(s) \mathbb{E}_{a \sim \mu_0^*}[q_{\mu^*,0}^c(s, a)]$$

$$\leq \sum_s p_0(s) \delta_{\epsilon,0}(s) \qquad \text{(Theorem 1)}$$

$$= \sum_s p_0(s)(1 + \epsilon) v_{\pi,0}^c(s) \qquad \text{(By (13))}$$

$$= (1 + \epsilon) \sum_s p_0(s) v_{\pi,0}^c(s)$$

$$= (1 + \epsilon) J^c(\pi),$$

which completes the proof. $\qquad \square$

### A.7 PROOF OF LEMMA 5

*Proof.* $\forall s, a$, when $t = T - 1$, $\tilde{r}_t(s, a) = r_{\pi,t}(s, a)^2$, as defined in (14). For $t \in [T - 2]$,

$$\tilde{r}_t(s, a)$$

$$= \nu_{\pi,t}(s, a) + q_{\pi,t}(s, a)^2 + \mathbb{E}_{S_{t+1}}\left[\mathbb{V}\left(G^{\text{PDIS}}(\tau_{t+1:T-1}^{\mu_{t+1:T-1}^*}) \mid S_{t+1}\right) \mid s, a\right] \qquad \text{(By (14))}$$

$$= \nu_{\pi,t}(s, a) + q_{\pi,t}(s, a)^2$$

$$\quad + \sum_{s'} p(s'|s, a)\left[\mathbb{E}_{A_{t+1} \sim \mu_{t+1}^*}\left[\rho_{t+1}^2\left(\mathbb{E}_{S_{t+2}}\left[\mathbb{V}\left(G^{\text{PDIS}}(\tau_{t+2:T-1}^{\mu_{t+2:T-1}^*}) \mid S_{t+2}\right) \mid S_{t+1}, A_{t+1}\right]\right.\right.$$

$$\quad + \nu_{\pi,t+1}(S_{t+1}, A_{t+1}) + q_{\pi,t+1}(S_{t+1}, A_{t+1})^2) \mid S_{t+1} = s'] - v_{\pi,t+1}(s')^2] \qquad \text{(By Lemma 6)}$$

$$= \nu_{\pi,t}(s, a) + q_{\pi,t}(s, a)^2 + \sum_{s'} p(s'|s, a)\left[\mathbb{E}_{A_{t+1} \sim \mu_{t+1}^*}\left[\rho_{t+1}^2 \tilde{r}_{\pi,t+1}(S_{t+1}, A_{t+1}) \mid S_{t+1} = s'\right]\right.$$

$$\quad - v_{\pi,t+1}(s')^2] \qquad \text{(By (14))}$$

$$= \nu_{\pi,t}(s,a) + q_{\pi,t}(s,a)^2 + \sum_{s',a'} p(s'|s,a)\big[\rho_{t+1}\pi_{t+1}(a'|s')\tilde{r}_{\pi,t+1}(s',a') - v_{\pi,t+1}(s')^2\big].$$

$$= \mathbb{V}_{S_{t+1}}\left(v_{\pi,t+1}(S_{t+1}) \mid S_t = s, A_t = a\right) + q_{\pi,t}(s,a)^2$$
$$\qquad + \sum_{s',a'} p(s'|s,a)\big[\rho_{t+1}\pi_{t+1}(a'|s')\tilde{r}_{\pi,t+1}(s',a') - v_{\pi,t+1}(s')^2\big] \qquad \text{(Definition of } \nu)$$

$$= \mathbb{E}_{S_{t+1}}\left[v_{\pi,t+1}(S_{t+1})^2 \mid S_t = s, A_t = a\right] - \mathbb{E}_{S_{t+1}}\left[v_{\pi,t+1}(S_{t+1}) \mid S_t = s, A_t = a\right]^2$$
$$\qquad + q_{\pi,t}(s,a)^2 + \sum_{s',a'} p(s'|s,a)\big[\rho_{t+1}\pi_{t+1}(a'|s')\tilde{r}_{\pi,t+1}(s',a') - v_{\pi,t+1}(s')^2\big]$$

$$= \sum_{s'} p(s'|s,a)v_{\pi,t+1}(s')^2 - (q_{\pi,t}(s,a) - r(s,a))^2 + q_{\pi,t}(s,a)^2$$
$$\qquad + \sum_{s',a'} p(s'|s,a)\rho_{t+1}\pi_{t+1}(a'|s')\tilde{r}_{\pi,t+1}(s',a') - \sum_{s'} p(s'|s,a)v_{\pi,t+1}(s')^2$$

$$= 2q_{\pi,t}(s,a)r(s,a) - r(s,a)^2 + \sum_{s',a'} p(s'|s,a)\rho_{t+1}\pi_{t+1}(a'|s')\tilde{r}_{\pi,t+1}(s',a')$$

$$= 2q_{\pi,t}(s,a)r(s,a) - r(s,a)^2 + \mathbb{E}_{s'\sim p, a'\sim\pi}\left[\frac{\pi_{t+1}(a'|s')}{\mu^*_{t+1}(a'|s')}\tilde{r}_{\pi,t+1}(s',a')\right].$$

$$\square$$

## B EXPERIMENT DETAILS

### B.1 GRIDWORLD

| Environment Size | On-policy MC | **Ours** | ODI | ROS |
|---|---|---|---|---|
| 1,000 | 1.000 | **0.547** | 0.460 | 0.953 |
| 27,000 | 1.000 | **0.575** | 0.484 | 0.987 |

Table 3: Relative variance for estimators on Gridworld. The relative variance is defined as the variance of each estimator divided by the variance of the on-policy Monte Carlo estimator. Numbers are averaged over 900 independent runs (30 target policies, each having 30 independent runs). Standard errors are plotted in Figure 1.

| Env Size | On-policy MC | **Ours** | ODI | ROS | Saved Cost Percentage |
|---|---|---|---|---|---|
| 10 | 1000 | **472** | 738 | 1035 | (1000 - 472)/1000 = **52.8%** |
| 30 | 1000 | **487** | 765 | 1049 | (1000 - 487)/1000 = **51.3%** |

Table 4: Cost needed to achieve the same estimation accuracy that on-policy Monte Carlo achieves with 1000 episodes on Gridworld. Each number is averaged over 900 runs. Standard errors are plotted in Figure 2.

We conduct experiments on Gridworlds with $n^3 = 1,000$ and $n^3 = 27,000$ states, where for a Gridworld with size $n^3$, we set the width, height, and time horizon $T$ all to be $n$. The action space contains four different possible actions: up, down, left, and right. After taking an action, the agent has a probability of 0.9 to move accordingly and a probability of 0.1 to move uniformly at random. When the agent runs into a boundary, it stays in its current position. We randomly generate the reward function $r(s,a)$ and cost function $c(s,a)$. We consider 30 randomly generated target policies with various performances. The ground truth policy performance is estimated by the on-policy Monte Carlo method, running each target policy for $10^6$ episodes. We experiment two different sizes of the Gridworld with a number of $1,000$ and $27,000$ states.

The offline dataset of each environment contains a total of $1,000$ episodes generated by 30 policies with various performances. The performance of those policies ranges from completely random

initialized policies to well-trained policies in each environment. For example, in Hopper, the performance of those 30 policies ranges from around 18 to around 2800. We let offline data be generated by various policies to simulate the fact that offline data are from different past collections.

We learn functions $q_{\pi,t}, q^c_{\pi,t}$, and $\hat{r}_{\pi,t}$ using Fitted Q-Evaluation algorithms (FQE, Le et al. (2019)) by passing data tuples in $\mathcal{D}_\nu$ from $t = T - 1$ to 0. It is worth noticing that Fitted Q-Evaluation (FQE, Le et al. (2019)) is a different algorithm from Fitted Q-Improvement (FQI). Importantly, Fitted Q-Evaluation is not prone to overestimate the action-value function $q_{\pi,t}$ because it does not have any max operator and does not change the policy. All hyperparameters are tuned offline based on Fitted Q-learning loss. We leverage a one-hidden-layer neural network and test the neural network size with $[64, 128, 256]$. We then choose 64 as the final size. We also test the learning rate for Adam optimizer with $[1e^{-5}, 1e^{-4}, 1e^{-3}, 1e^{-2}]$ and finally choose to use the default learning rate $1e^{-3}$ as learning rate for Adam optimizer (Kingma and Ba, 2015). For the benchmark algorithms, we use their reported hyperparameters (Zhong et al., 2022; Liu and Zhang, 2024). Each policy has 30 independent runs, resulting in a total of $30 \times 30 = 900$ runs. Thus, each curve in Figure 1, Figure 2 and each number in Table 1, Table 3 and Table 4 are averaged from 900 different runs over a wide range of policies, demonstrating a strong statistical significance.

## B.2 MuJoCo

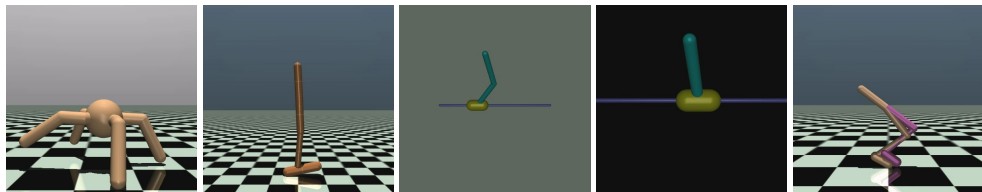

Figure 4: MuJoCo robot simulation tasks (Todorov et al., 2012). Pictures are adapted from (Liu and Zhang, 2024). Environments from the left to the right are Ant, Hopper, InvertedDoublePendulum, InvertedPendulum, and Walker.

| | On-policy MC | Ours | ODI | ROS |
|---|---|---|---|---|
| Ant | 1.000 | **0.835** | 0.811 | 1.032 |
| Hopper | 1.000 | **0.596** | 0.542 | 1.005 |
| I. D. Pendulum | 1.000 | **0.778** | 0.724 | 0.992 |
| I. Pendulum | 1.000 | **0.439** | 0.351 | 0.900 |
| Walker | 1.000 | **0.728** | 0.696 | 0.908 |

Table 5: Relative variance of estimators on MuJoCo. The relative variance is defined as the variance of each estimator divided by the variance of the on-policy Monte Carlo estimator. All numbers are averaged over 900 independent runs (30 target policies, each having 30 independent runs).

| | On-policy MC | Ours | ODI | ROS |
|---|---|---|---|---|
| Ant | 1.000 | **0.897** | 1.397 | 1.033 |
| Hopper | 1.000 | **0.930** | 1.523 | 1.021 |
| I. D. Pendulum | 1.000 | **0.876** | 1.399 | 1.012 |
| I. Pendulum | 1.000 | **0.961** | 1.743 | 0.990 |
| Walker | 1.000 | **0.953** | 1.485 | 1.061 |

Table 6: Average trajectory cost on MuJoCo. Numbers are normalized by the cost of the on-policy estimator. ODI and ROS have much larger costs because they both ignore safety constraints. **Our method is the only method consistently achieving both variance reduction and safety constraint satisfaction.**

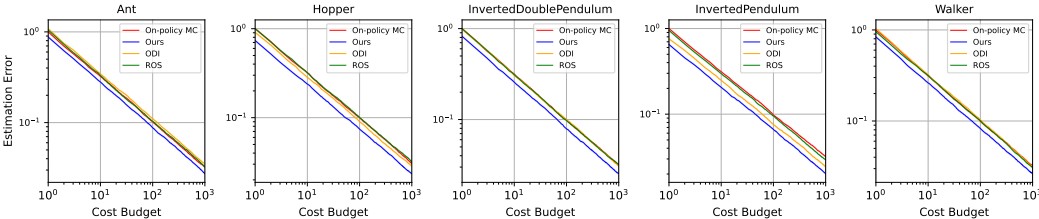

Figure 5: Results on MuJoCo with log-scale y-axis to show the error does not converge. Each curve is averaged over 900 runs (30 target policies, each having 30 independent runs). Shaded regions denote standard errors and are invisible for some curves because they are too small.

MuJoCo is a physics engine with various stochastic environments, in which the goal is to control a robot to achieve different behaviors such as walking, jumping, and balancing. We construct 30 policies in each environment, resulting a total of 150 policies. The policies demonstrate a wide range of performance generated by the proximal policy optimization (PPO) algorithm (Schulman et al., 2017) using the default PPO implementation in Huang et al. (2022). Original MuJoCo environments are Markov decision processes (MDP) and do not have cost functions. We enhance it with cost functions to make it constrained Markov decision processes (CMDP). Specifically, the cost of the MuJoCo environments is built on the control cost of the robot. The control cost is the L2 norm of the action and is proposed by OpenAI Gymnasium (Brockman et al., 2016). This control cost is motivated by the fact that large actions in robots induce sudden changes in the robot's state and may cause safety issues.

We set each environment in MuJuCo to have a fixed time horizon 100 in OpenAI Gymnasium (Towers et al., 2024). Because our methods are designed for discrete action space, we discretize the first dimension of the MuJoCo action space. The remaining dimensions are then controlled by the PPO policies and are deemed as part of the environment. The offline dataset for each environment contains 1,000 episodes generated by 30 policies with various performances, following the same method as in the Gridworld environments. Functions $q_{\pi,t}$, $q_{\pi,t}^c$, and $\hat{r}_{\pi,t}$ are learned using the same way as in Gridworld environments. Notably, our algorithm is robust on hyperparameters, as all hyperparameters in Algorithm 1 are tuned offline and are the same across all MuJoCo and Gridworld experiments. Each policy in MuJoCo has 30 independent runs, resulting in a total of $30 \times 30 = 900$ runs. As a result, curves in all figures are averaged from 900 different runs with a wide range of policies, showing a strong statistical significance.

