# OpenReview forum: "Efficient Policy Evaluation with Safety Constraint for Reinforcement Learning"
_ICLR.cc/2025/Conference — ICLR 2025 Poster_

### Official Review · Reviewer_MExc · 2024-10-16

**Soundness:** 3
**Presentation:** 3
**Contribution:** 3
**Rating:** 8
**Confidence:** 5

**Summary:**

This paper proposes an optimal variance-minimizing behavior policy to guarantee the satisfaction of safety constraints. Theoretically, the authors prove that their proposed method is unbiased and with lower variance. Empirical experiments show that their proposed method achieves variance reduction and safety guarantee.

**Strengths:**

- The authors address a very important and interesting problem where safety constraint is considered in RL policy evaluation.
- The motivation behind the problem formulation is well-presented while using an example of Google's data center.
- The background section is well-written and easy to follow.
- I think it is a good idea to provide theoretical results and key ideas in the case of contextual bandit.
- Theoretical results are nice and proofs are well-presented.
- The empirical experiments are overall well-conducted.
- The authors' proposed method performs well in Grid-world and MuJoCO.

**Weaknesses:**

- In the Related Work section, the literature review of safe RL can be written in a more organized way. There are several survey papers on safe RL, so the authors may want to refer to how to comprehend safe RL.
    - Brunke, Lukas, et al. "Safe learning in robotics: From learning-based control to safe reinforcement learning." Annual Review of Control, Robotics, and Autonomous Systems 5.1 (2022): 411-444.
    - Liu, Yongshuai, Avishai Halev, and Xin Liu. "Policy learning with constraints in model-free reinforcement learning: A survey." In IJCAI. 2021.
    - Gu, Shangding, et al. "A review of safe reinforcement learning: Methods, theory and applications." arXiv preprint arXiv:2205.10330 (2022).
    - Wachi, Akifumi, Xun Shen, and Yanan Sui. "A Survey of Constraint Formulations in Safe Reinforcement Learning." In IJCAI (2024)
- Readers cannot know that this paper deals with off-policy evaluation until line 160. Please clearly mention that in the Abstract or Introduction. It may be better to add "off-policy" in the title.
- I think lines 309-369 are hard to follow. I understand that (12) is challenging to handle, but it is unclear whether the authors' transformation is reasonable or not. I think it is better to discuss the theoretical relations between (12) and (15, 16), e.g., equivalent, conservative approximation, etc.
- I think readers cannot identify the magnitude of variance for each algorithm from Figures 1 or 2. There should be a table including the actual variance numbers (at least in the Appendix) like Tables 3 or 4.
- I am also wondering whether the variance of on-policy MC in Tables 3 or 4 is large. I agree with the authors that the proposed method achieves lower variance, but I am not fully convinced of how important are the differences in real applications.
- I think the reproducibility of this paper is rather low given there is no source code attached and there is little information on the experimental setups. It is problematic as a research paper that even basic information is not mentioned for reproducing experimental results.

### Typos
- In (3), $\mathbb{V}$ --> $\mathbb{V}_{a \sim \mu}$.

**Questions:**

- Q1: I could not understand (12). Why is the safety threshold $(1+ \epsilon)$? In other words, could you tell me the reason why RHS in (12) does not depend on $\pi$? Perhaps is this a typo?
- Q2: The original MuJoCO environment does not have a notion of safety cost. How did the authors introduce safety costs?
- Q3: How was the offline dataset generated? Could you tell me the details? I know there is some statement in the Appendix, but it is critical for understanding the actual performance of the algorithm.
- Q4: Could you show me the experimental results or figures illustrating how the performance of each algorithm is affected by the number of offline datasets?

---

> ### Author Response · Authors · 2024-11-20
>
> Thank you for your extremely detailed review and practical suggestions. Your comments show that our work is well written, and the variance reduction and safety compliance of our method is guaranteed both theoretically and empirically.
>
> ### For weaknesses:
> > There are several survey papers on safe RL, so the authors may want to refer to how to comprehend safe RL.
>
> Thank you for your practical suggestion!  We have adjusted our related work section and cited these papers in our main text (line 69-70 and line 81-83).
>
> >Readers cannot know that this paper deals with off-policy evaluation until line 160. Please clearly mention that in the Abstract or Introduction. It may be better to add "off-policy" in the title.
>
> Many thanks for this practical suggestion. We have taken it to heart! We have added “off-policy” into our title for better understanding.
>
> > I think it is better to discuss the theoretical relations between (12) and (15, 16), e.g., equivalent, conservative approximation, etc.
>
> We have taken this suggestion to heart. In (12), we require that $\quad J^c(\mu^*)\leq (1+\epsilon)J^{c}(\pi)$, where $J^c(\mu^*)$ and $J^{c}(\pi)$ are the expected total cost of policy $\mu^*$ and the target policy $\pi$, respectively. In (16), we further require that $E_{a\sim\mu_t}[q^c_{\mu,t}(s,a)]\leq (1+\epsilon)E_{a\sim\mu_t}[q^c_{\mu,t}(s,a)]$ for all time steps $t$. In fact, the constraint in (16) is **stricter** than (12). In Theorem 3, we proved that our behavior policy $\mu^*$ derived under (15)(16) does satisfy the wider constraint (12).
>
> We have added more discussions on the theoretical relations between than in our main text (line 369-370). Thanks again for your practical suggestion!
>
> >I think readers cannot identify the magnitude of variance for each algorithm from Figures 1 or 2. There should be a table including the actual variance numbers (at least in the Appendix) like Tables 3 or 4.
>
> Thank you for pointing this out! We agree that including a table with the variance and cost reduction numbers would provide a clearer representation of the results. To address this, we have added Table 3 and 4 (corresponding to Figure 1 and 2, respectively) to the appendix and attached them below:
> |Environment Size | On-policy MC | **Ours** | ODI  | ROS|
> |------------------|--------------|----------|-------|-------|
> | 1,000            | 1.000        | **0.547** | 0.460 | 0.953 |
> | 27,000           | 1.000        | **0.575** | 0.484 | 0.987 |
>
> *Table 3: Relative variance for estimators on Gridworld. Each number is averaged over 900 runs. Standard errors are plotted in Figure 1.*
>
> | Env Size | On-policy MC | **Ours** | ODI  | ROS   | Saved Cost Percentage |
> |----------|--------------|----------|-------|-------|------------------------|
> | 1,000       | 1000         | **472**  | 738   | 1035  | **(1000 - 472)/1000 = 52.8%** |
> | 27,000       | 1000         | **487**  | 765   | 1049  | **(1000 - 487)/1000 = 51.3%** |
>
> *Table 4: Cost needed to achieve the same estimation accuracy that on-policy Monte Carlo achieves with 1000 episodes on Gridworld. Each number is averaged over 900 runs. Standard errors are plotted in Figure 2.*
>
> Besides, the numbers of *average trajectory cost* on Gridworld have been shown in our Table 1.
> Considering *solely* variance reduction, Table 3 shows that our method achieves greatly lower variance than the on-policy MC method. In Table 4, we further demonstrate that our method **saves more than 50% of cost to achieve the desired estimation accuracy**, outperforming all three baselines significantly.

---

> ### Author Response · Authors · 2024-11-20
>
> >I am also wondering whether the variance of on-policy MC in Tables 3 or 4 is large. I agree with the authors that the proposed method achieves lower variance, but I am not fully convinced of how important are the differences in real applications.
>
> **-- *Safety* constrained variance reduction**
>
> In fact, our algorithm does not consider *solely* variance reduction, but instead addresses **variance reduction and safety compliance simultaneously**. As pointed out in our experiment section with data from Table 1,2 and 6, solely reducing variance usually comes with a **trade-off of a higher cost**. Thus, we believe that our method should be evaluated not only by how much variance it reduces, but also how much cost it saves to achieve the desired evaluation accuracy.
>
> As shown in *Table 2* of our paper, our method achieves state-of-the-art performance in **reducing the cost needed to achieve the desired evaluation accuracy**. Specifically, it saves from **$25.4$% to $57.5$%** cost across different MuJoCo environments. This superior performance is a result of the **joint effects of reducing both variance and controlling cost**, as respectively demonstrated in Table 3 and 4 (now Table 5 and 6 in our revision). Besides, all our numbers are averaged over $900$ different runs over a wide range of policies, indicating strong statistical significance.
>
>
> | Environment       | On-policy MC | **Ours** | ODI [1]   | ROS [2] | Saved Cost (%) |
> |-------------------|--------------|----------|-------|-------|----------------|
> | Ant              | 1000         | 746      | 1136  | 1063  | **25.4%**      |
> | Hopper           | 1000         | 552      | 824   | 1026  | **44.8%**      |
> | I. D. Pendulum   | 1000         | 681      | 1014  | 1003  | **31.9%**      |
> | I. Pendulum      | 1000         | 425      | 615   | 890   | **57.5%**      |
> | Walker           | 1000         | 694      | 1031  | 960   | **30.6%**      |
>
> *Table 2:Cost needed to achieve the same estimation accuracy that on-policy Monte Carlo achieves with $1000$ episodes on MuJoCo. Each number is averaged over 900 independent  runs. Standard errors are plotted in Figure 3.*
>
> >I think the reproducibility of this paper is rather low given there is no source code attached and there is little information on the experimental setups.
>
> Thank you for pointing this out! We will publish the source code upon publication to facilitate future research.
> We have also included more explanations on experimental setups in Appendix B (line 1078-1082) of the revision. Additional details on experiments are provided in our answer to your Q2-4.
>
> ### For Typos:
> >In (3), $V\to V_{ a \sim u}$
>
> Many thanks for this extremely detailed notification! We added this subscript into (3) for clarification.
>
> ### For questions:
>
> Q1
>
> >In other words, could you tell me the reason why RHS in (12) does not depend on $\pi$? Perhaps is this a typo?
>
> Thank you for catching this typo. You are correct that the RHS of equation (12) should depend on $\pi$, and it is actually $(1+\epsilon)J^c(\pi)$, where $J^c(\pi)$ is the expected cost of the target policy. We have corrected this oversight in the current paper.
>
> Indeed, we have noticed this typo and corrected them right after the submission. We have also once again checked our work thoroughly to avoid any typo.
>
> Q2
> > The original MuJoCO environment does not have a notion of safety cost. How did the authors introduce safety costs?
>
> The cost of the MuJoCo environments is built on **the control cost of the robot**. The control cost is the L2 norm of the action and is proposed by OpenAI Gymnasium (Brockman et al., 2016). This control cost is motivated by the fact that large actions in robots induce sudden changes in the robot's state and may cause safety issues.
>
> Q3
> >How was the offline dataset generated? Could you tell me the details? I know there is some statement in the Appendix, but it is critical for understanding the actual performance of the algorithm.
>
>
> We are glad to provide more details! The offline dataset of each environment contains a total of 1, 000 episodes generated by
> 30 policies with various performances. The performance of those policies **ranges from completely
> random initialized policies to well-trained policies** in each environment. For example, in the Openai Gymnasium Hopper-v4, the performance of those 30 policies ranges from around 18 to around 2800. We let offline data be
> generated by various policies to **simulate the fact that offline data are from different past collections**.

---

> > ### Author Response · Authors · 2024-11-20
> >
> > Q4
> > >Could you show me the experimental results or figures illustrating how the performance of each algorithm is affected by the number of offline datasets?
> >
> > To better answer your question, we provide the result of ablation studies using **different numbers of offline data**. In the tables below, 1,2,3 K means we use offline data with $1000$, $2000$ and $3000$ episodes, respectively. Also, notice that SCOPE (Safety-Constrained Off-Policy Evaluation) is the name of our method.
> > |               | On-policy MC | SCOPE-1K | SCOPE-2K | SCOPE-3K | Saved Cost Percentage |
> > |---------------|--------------|-----------|-----------|-----------|------------------------|
> > | Ant       | 1000         | 746       | 707       | 687       | **25.4% – 31.3%**     |
> > | Hopper    | 1000         | 552       | 515       | 488       | **44.8% – 51.2%**     |
> > | I. D. Pendulum | 1000    | 681       | 641       | 621       | **31.9% – 37.9%**     |
> > | I. Pendulum    | 1000    | 425       | 388       | 369       | **57.5% – 63.1%**     |
> > | Walker    | 1000         | 694       | 667       | 647       | **30.6% – 35.3%**     |
> >
> > *Table 7: Cost needed to achieve the same estimation accuracy that on-policy Monte Carlo achieves
> > with 1000 online episodes on MuJoCo. Each number is averaged over 900 independent  runs.*
> >
> > |        | On-policy MC | SCOPE-1K | SCOPE-2K | SCOPE-3K |
> > |----------|--------------|-----------|-----------|-----------|
> > | Ant       | 1.000        | 0.835     | 0.809     | 0.780     |
> > | Hopper    | 1.000        | 0.596     | 0.564     | 0.531     |
> > | I. D. Pendulum | 1.000   | 0.778     | 0.730     | 0.718     |
> > | I. Pendulum    | 1.000   | 0.439     | 0.401     | 0.389     |
> > | Walker    | 1.000        | 0.728     | 0.709     | 0.690     |
> >
> > *Table 8: Relative variance of estimators on MuJoCo. The relative variance is defined as the variance
> > of each estimator divided by the variance of the on-policy Monte Carlo estimator. Each number is averaged over 900 independent  runs.*
> >
> >
> > |               | On-policy MC | SCOPE-1K | SCOPE-2K | SCOPE-3K |
> > |---------------|--------------|-----------|-----------|-----------|
> > | Ant       | 1.000        | 0.897     | 0.881     | 0.877     |
> > | Hopper    | 1.000        | 0.930     | 0.921     | 0.918     |
> > | I. D. Pendulum | 1.000   | 0.876     | 0.874     | 0.867     |
> > | I. Pendulum    | 1.000   | 0.961     | 0.958     | 0.956     |
> > | Walker | 1.000        | 0.953     | 0.949     | 0.946     |
> >
> > *Table 9: Average trajectory cost on MuJuCo under different . Numbers are normalized by the cost of the on-policy estimator.  Each number is averaged over 900 independent runs.*
> >
> > As shown in the above tables, our method scales with the number of offline data. Specifically, in Table 7, we **saved 25.4%-63.1\% cost** in achieving the desired estimation accuracy across different environments and different offline data numbers, compared with the on-policy Monte Carlo method. Besides, in Table 8 & Table 9, we see that both the variance and the average trajectory cost are reduced.
> >
> >
> > [1] (Liu and Zhang, ICML 2024) “Efficient Policy Evaluation with Offline Data Informed Behavior Policy Design” (ODI)
> >
> > [2] (Zhong et al. NeurIPS 2022) "Robust On-Policy Sampling for Data-Efficient Policy Evaluation in Reinforcement Learning" (ROS)

---

> > > ### Comment · Reviewer_MExc · 2024-11-20
> > > **Thank you for clarifications**
> > >
> > > I appreciate the authors for their rebuttals and additional experiments. My concerns have been addressed. If Eq. (12) is a typo, then I am fully convinced by the claims of this paper. I increased my score to 8 and confidence to 5.

---

> > > > ### Author Response · Authors · 2024-11-20
> > > >
> > > > Perfect, thanks again for the extremely constructive comments and the fast turn around.

---

### Official Review · Reviewer_J4Sc · 2024-10-27

**Soundness:** 3
**Presentation:** 2
**Contribution:** 2
**Rating:** 6
**Confidence:** 4

**Summary:**

This paper studies the problem of policy evaluation under constraints and proposes an algorithm that reduces the policy evaluation variance while satisfying the constraint.

**Strengths:**

The proposed solution is derived by formulating and solving an optimization problem, which has support from the theoretical results.

The literature review is thoughtful and complete.

**Weaknesses:**

In the experiment, the performances of the constrained RL algorithm under different policy evaluation methods should be compared.

See **Questions**

**Questions:**

Is the transition from (12) to (13) too conservative? As (12) requires the total cost to be smaller than the threshold but (13) makes the feasible set smaller. What if the optimization problem (13) has an empty feasible set? How to derive Theorem 3 when (13) is too conservative, as the policy $\pi$ will not satisfy this conservative constraint?

How is the proposed method applied to continuous state-action space?

The mainstream policy evaluation uses a one-step method, such as the TD method. Can the proposed method be applied to these methods?

---

> ### Author Response · Authors · 2024-11-20
>
> Thank you so much for your time and comments. As you pointed out, our method is theoretically grounded, and our work also includes a thoughtful and comprehensive literature review.
>
> ### For weaknesses:
>
> >In the experiment, the performances of the constrained RL algorithm under different policy evaluation methods should be compared.
>
> Thank you for your insightful comment! We have two interpretations for this question:
>
> **1. Application of our method under other policy evaluation methods (TD):**
>
> As answered in your last question, we believe that in online policy evaluation, the on-policy Monte Carlo method is the standard approach. It provides an **unbiased scalar performance metric** that every RL implementation requires (i.e., for hyperparameter tuning or for model selection). By contrast, TD is **biased because it needs bootstrapping and function approximation**. So when practitioners need an unbiased scalar performance metric, MC is the dominating method [1][2].
>
> Additionally, the well-known ICML paper (Jiang and Li, 2016,
>  *Doubly Robust Off-Policy Value Evaluation For Reinforcement Learning* ) as well as [1][2] are all different Monte Carlo estimators. Therefore, in this work, we also develop our algorithm using the Monte Carlo method, and use different Monte Carlo estimators as baselines.
>
> Besides, as specified in our conclusion (line 539), extending our constrained variance minimization technique to the TD method is our future work.
>
> **2. Comparison of our method with other safety constrained RL algorithm:**
>
> Thank you for pointing this out. In fact, compared with mainstream safe RL papers focusing on policy improvement, our work addresses the safety in policy evaluation. Specifically, we aim at reducing the evaluation variance while satisfying safety constraints.
>
> However, to the best of our knowledge, [4] is the only existing work considering **safety-constrained policy evaluation** in reinforcement learning. However, while our method and the baselines we compared [1][2] are designed for **general MDPs** and are **model-free**, [4], as suggested by its title, focuses solely on **tabular MDPs** and is **model-based**. It is not clear to us how [4] can be used in our MuJoCo experiments, which are non tabular. Given its limited applicability, we have not included [4] as a baseline in our comparisons.
>
>
> Please let us know if we have addressed your comments! If there are methods that you want us to compare with, could you please refer us to them? We would be happy to discuss.
>
>
> ### For questions:
>
> >Is the transition from (12) to (13) too conservative? As (12) requires the total cost to be smaller than the threshold but (13) makes the feasible set smaller. What if the optimization problem (13) has an empty feasible set? How to derive Theorem 3 when (13) is too conservative, as the policy $\pi$ will not satisfy this conservative constraint?
>
> Thanks for your comment! We acknowledge that there was a typo in (12), which we have fixed in our revision pdf. In (12), we require that $\quad J^c(\mu^*)\leq (1+\epsilon)J^{c}(\pi)$, where $J^c(\mu^*)$ and $J^{c}(\pi)$ are the expected total cost of policy $\mu^*$ and the target policy $\pi$, respectively.
>
> While in (13), we further require that $ E_{a\sim\mu_t}[q^c_{\mu,t}(s,a)]\leq (1+\epsilon) E_{a\sim\mu_t}[q^c_{\mu,t}(s,a)]$ for all time steps $t$. The constraint in (13) is a **sufficient condition** for (12), as it ensures that our behavior policy $\mu$ performs under the safety threshold throughout the time steps. In Theorem 3, we proved that our derived behavior policy $\mu^*$ (with constraint (13)) **does satisfy the original and wider constraint (12)**.
>
> For your feasibility concern, we have proved in Appendix A.4 that **$\pi$ is in the feasible set under the constraint (13)**. Please let us know if you have any further question. We are happy to discuss!
>
>
> >How is the proposed method applied to continuous state-action space?
>
> Our method is applicable to continuous state space with discrete action space. Specifically, we estimate the $\tilde{r}$ functions by passing the continuous states into the neuro network and getting the value for each action. We then use $\tilde{r}$ to construct our behavior policy $\mu^*$ by (15) for safe online data collecting.

---

> ### Author Response · Authors · 2024-11-20
>
> >The mainstream policy evaluation uses a one-step method, such as the TD method. Can the proposed method be applied to these methods?
>
> Yes, our method has the potential to be applied to the TD method. Specifically, our safety-constrained variance reduction problem formulation has the potential to be applied to one-step data collection in TD. As mentioned at the end of our conclusion section, this is one future direction of our work.
>
> Besides, we believe that in online policy evaluation, the on-policy Monte Carlo method is the standard approach. E.g., when an RL practitioner wants to draw a curve of their agents’ performance against training steps (we believe every empirical RL paper has such a curve), they will use Monte Carlo. The reason is that for this purpose (i.e., for hyperparameter tuning or for model selection), we need an unbiased **scalar performance metric**. TD is a bootstrapping method, which is more used for getting the value function. In general, **TD method is biased because it needs function approximation and bootstrapping**. So when practitioners **need an unbiased scalar performance metric**, MC is the dominating method [1][2]. This fact motivates us to propose an optimal variance-minimizing behavior policy under safety constraints for MC methods.
>
> Thank you again for your insightful question!
>
>
>
> [1] (ICML Liu and Zhang, 2024) “Efficient Policy Evaluation with Offline Data Informed Behavior Policy Design” (ODI)
>
> [2] (NeurIPS Zhong et al. 2022) "Robust On-Policy Sampling for Data-Efficient Policy Evaluation in Reinforcement Learning" (ROS)

---

> ### Author Response · Authors · 2024-11-25
>
> As the rebuttal phase is nearing its end, we wanted to kindly follow up to check if you had any additional feedback or comments for our paper. Your input would be greatly appreciated, and we are confident to discuss and address any concerns you may have.
>
> Thank you again for your time and effort in reviewing our work!

---

> > ### Comment · Reviewer_J4Sc · 2024-11-26
> >
> > Thanks for the response. My concerns have been addressed. I have adjusted my rating accordingly.

---

### Official Review · Reviewer_HUVe · 2024-11-04

**Soundness:** 3
**Presentation:** 3
**Contribution:** 2
**Rating:** 6
**Confidence:** 3

**Summary:**

This paper addresses high variance in on-policy evaluation in reinforcement learning by proposing a behavior policy that minimizes variance under safety constraints. Unlike previous methods that ignore safety, this approach provides unbiased, low-variance evaluation while ensuring safe execution. Empirical results show it outperforms prior methods in both variance reduction and safety compliance.

**Strengths:**

1. The study addresses the new and intriguing problem of incorporating safety constraints in policy evaluation.
2. The paper is well-organized and easy to follow.
3. It provides comprehensive support with both theoretical and empirical evidence.

**Weaknesses:**

1. While addressing safety constraints is novel, the approach to variance minimization under these constraints lacks significant innovation.
2. In the experimental section, the paper could benefit from comparisons with a broader range of work in safe reinforcement learning to better demonstrate its advantages.

**Questions:**

1. From my understanding, the algorithm proposed in this paper aims to learn a behavior policy that minimizes variance while satisfying safety constraints. However, I don’t see a clear connection in the theoretical results to the offline dataset used. Could you clarify this? Specifically, if the dataset lacks sufficient coverage, is it still possible to obtain a reliable behavior policy?

2. In the experimental section, the paper compares the proposed method to the Monte Carlo (MC) approach, which seems straightforward but rather basic. Could the authors also consider comparing with online bootstrapping methods, as they might provide more efficient solutions for online policy evaluation?

**Details Of Ethics Concerns:**

No concerns.

---

> ### Author Response · Authors · 2024-11-20
>
> Thanks a lot for your detailed comments. Your opinion shows that our paper provides solid theoretical foundations and empirical results, outperforming prior methods in both variance reduction and safety compliance.
>
> ### For weaknesses:
> >While addressing safety constraints is novel, the approach to variance minimization under these constraints lacks significant innovation.
>
> As pointed out by your review, previous methods published on ICML and NeurIPS [1][2][3]  only consider variance reduction, completely ignoring safety issues. Thus, although their methods achieve reduced variance, they indeed *increase the execution cost* compared with the on-policy MC method, as shown in Table 6 of our paper:
>
> |               | On-policy MC | **Ours** | ODI [1]   | ROS [2]  |
> |---------------|--------------|----------|-------|-------|
> | Ant       | 1.000        | **0.897** | 1.397 | 1.033 |
> | Hopper    | 1.000        | **0.930** | 1.523 | 1.021 |
> | I. D. Pendulum | 1.000   | **0.876** | 1.399 | 1.012 |
> | I. Pendulum    | 1.000   | **0.961** | 1.743 | 0.990 |
> |Walker    | 1.000        | **0.953** | 1.485 | 1.061 |
>
> *Table 6: Average trajectory cost on MuJoCo. Numbers are normalized by the cost of the on-policy estimator. ODI and ROS have much larger costs because they both ignore safety constraints.*
>
>
> By contrast, our method innovatively considers variance reduction and safety constraint satisfaction simultaneously.  In fact, our method is the **only method** to achieve both variance reduction and safety constraint satisfaction, compared with [1][2]. As computed from Table 2, our method saves up to $57.5$% cost to achieve the desired evaluation accuracy, surpassing all the baseline methods across various environments.
>
> |      | On-policy MC | **Ours** | ODI [1]   | ROS [2] | Saved Cost (%) |
> |-------------------|--------------|----------|-------|-------|----------------|
> | Ant              | 1000         | 746      | 1136  | 1063  | **25.4%**      |
> | Hopper           | 1000         | 552      | 824   | 1026  | **44.8%**      |
> | I. D. Pendulum   | 1000         | 681      | 1014  | 1003  | **31.9%**      |
> | I. Pendulum      | 1000         | 425      | 615   | 890   | **57.5%**      |
> | Walker           | 1000         | 694      | 1031  | 960   | **30.6%**      |
>
> *Table 2:Cost needed to achieve the same estimation accuracy that on-policy Monte Carlo achieves with $1000$ episodes on MuJoCo. Each number is averaged over 900 runs. Standard errors are plotted in Figure 3 of our paper.*
>
> >In the experimental section, the paper could benefit from comparisons with a broader range of work in safe reinforcement learning to better demonstrate its advantages.
>
> Thank you for pointing this out. In fact, compared with mainstream safe RL papers focusing on policy improvement, our work addresses the safety in policy evaluation. Specifically, we aim at reducing the evaluation variance while satisfying safety constraints.
>
> However, to the best of our knowledge, [4] is the only existing work considering **safety-constrained policy evaluation** in reinforcement learning. However, while our method and the baselines we compared [1][2] are designed for **general MDPs** and are **model-free**, [4], as suggested by its title, focuses solely on **tabular MDPs** and is **model-based**. It is not clear to us how [4] can be used in our MuJoCo experiments, which are non tabular. Given its limited applicability, we have not included [4] as a baseline in our comparisons.

---

> ### Author Response · Authors · 2024-11-20
>
> ### For questions:
>
> >Specifically, if the dataset lacks sufficient coverage, is it still possible to obtain a reliable behavior policy?
>
> Thank you for your question. As with most offline RL approaches, the number and coverage of offline data do impact the quality of the learned policy. Specifically, as we acknowledged in the conclusion of our paper, **there is no free lunch**: if the offline dataset contains only one data pair, it is clear that obtaining a reliable behavior policy is infeasible. This limitation is **inherent to offline RL** and cannot be fully resolved, as pointed out by Levein et al, 2020, “Offline Reinforcement Learning: Tutorial, Review, and Perspectives on Open Problems”.
>
>  > Could the authors also consider comparing with online bootstrapping methods, as they might provide more efficient solutions for online policy evaluation?
>
> Thank you for your insightful comment.
>
> We believe that in online policy evaluation, the on-policy Monte Carlo method is the standard approach. E.g., when an RL practitioner wants to draw a curve of their agents’ performance against training steps (we believe almost every RL paper has such a curve), they will just use Monte Carlo. The reason is that for this purpose (i.e., for hyperparameter tuning or for model selection), we need an unbiased **scalar performance metric**. If you refer to TD by bootstrapping method, we believe those bootstrapping methods are more used for getting the value function, not the scalar performance metric. In general, **those bootstrapping methods are biased because they need function approximation**. So when practitioners **need an unbiased scalar performance metric**, MC is the dominating method.
>
> Besides, the existing best-performing methods in policy evaluation [1][2][3] **all consider the on-policy MC as their primarily baseline**. Therefore, we also use the on-policy Monte Carlo and other Monte Carlo estimators as our baselines.
>
>
> [1] (ICML Jiang and Li, 2016) "Doubly Robust Off-Policy Value Evaluation For Reinforcement Learning"(DR)
>
> [2] (ICML Liu and Zhang, 2024) “Efficient Policy Evaluation with Offline Data Informed Behavior Policy Design” (ODI)
>
> [3] (NeurIPS Zhong et al. 2022) "Robust On-Policy Sampling for Data-Efficient Policy Evaluation in Reinforcement Learning" (ROS)
>
> [4](ICML Mukherjee et al.2024)"SaVeR: Optimal Data Collection Strategy for Safe Policy Evaluation in Tabular MDP”

---

> ### Author Response · Authors · 2024-11-25
>
> As the rebuttal phase is nearing its end, we wanted to kindly follow up to check if you had any additional feedback or comments for our paper. Your input would be greatly appreciated, and we are confident to discuss and address any concerns you may have.
>
> Thank you again for your time and effort in reviewing our work!

---

> ### Comment · Reviewer_HUVe · 2024-11-26
>
> I appreciate the authors for their rebuttals and explanations. My concerns have been addressed. I have increased my score to 6.

---

### Official Review · Reviewer_92mj · 2024-11-09

**Soundness:** 4
**Presentation:** 4
**Contribution:** 4
**Rating:** 8
**Confidence:** 2

**Summary:**

This paper provides an on-policy evaluation method that aims to reduce evaluation variance while also ensuring safety. This is done in the context of contextual bandits, sequential RL, and offline RL. Via theoretical results, this method is shown to be feasible, unbiased, and variance-reducing. Empirical results on GridWorld and MuJoCo demonstrate that under different cost budgets, the proposed method is able to improve performance (variance reduction) over baselines.

**Strengths:**

* This work is well-motivated in terms of aiming to reduce variance while also _ensuring safety_ during on-policy evaluation.
* The proposed method is simple, and theoretical results demonstrate that the method is feasible, unbiased, and variance-minimizing.
* The experimental demonstration is solid, with improved performance (variance reduction) over baselines under any given cost budget. The authors demonstrate good experimental practices, including averaging over many runs and comparing against strong baselines.

**Weaknesses:**

More discussion could be included on the specific form of the safety constraint. In what settings is it sufficient to ensure that "the expected cost of the designed behavior policy $\mu$ should be smaller than the multiple of the expected cost of the target policy $\pi$"? In what settings might this form of safety constraint be insufficient?

**Questions:**

* What are the implications of considering the undiscounted setting, as opposed to the discounted setting?
* See "Weaknesses"

---

> ### Author Response · Authors · 2024-11-20
>
> Many thanks for the encouraging feedback. Your comments truly highlight the core strength of this work: combining rigorous theoretical results with state-of-the-art empirical performance.
>
> ### For weaknesses:
>
> >More discussion could be included on the specific form of the safety constraint.
>
> In equation (18), we prove that our designed behavior policy $\mu^*$ satisfies the safety constraint $\quad J^c(\mu^*)\leq (1+\epsilon)J^{c}(\pi)$, where $J^c(\mu^*)$ and $J^{c}(\pi)$ are the expected total cost of policy $\mu^*$ and the target policy $\pi$, respectively. By setting $\epsilon=0$, as is the choice in our experiment section, **we aim to find a variance-reducing behavior policy without increasing the execution cost compared with the on-policy MC method**. In fact, under the threshold $\epsilon=0$, our method is the **only method** to achieve both variance reduction and safety constraint satisfaction, compared with the existing best-performing methods [1][2] published on ICML and NeurIPS.
>
> Besides, in situations where variance reduction becomes a priority, allowing $\epsilon$ to be slightly greater than $0$ can be a reasonable trade-off to achieve greater variance reduction.
>
>
> ### For questions:
> >What are the implications of considering the undiscounted setting, as opposed to the discounted setting?
>
> Thank you for your insightful question.
> Because we consider finite horizon MDPs, we use the undiscounted setting for simplifying notations. Our method can be directly extended to the discounted setting by simply adding the discount factor $\gamma$ into all derivations.
>
>
> [1] (Liu and Zhang, ICML 2024) “Efficient Policy Evaluation with Offline Data Informed Behavior Policy Design” (ODI)
>
> [2] (Zhong et al. NeurIPS 2022) "Robust On-Policy Sampling for Data-Efficient Policy Evaluation in Reinforcement Learning" (ROS)

---

### Meta-Review · Area_Chair_PUbt · 2024-12-21

**Metareview:**

The paper provides an on-policy evaluation method that aims to reduce the variance while satisfying safety constraints.
The problem is an important one and the reviewers are generally positive about the paper. The experiments are also well executed, and I encourage the authors to open source their code.

**Additional Comments On Reviewer Discussion:**

Minor concerns have been addressed during the rebuttal, and the reviewers are all positive about the paper

---

### Decision · Program_Chairs · 2025-01-22

Accept (Poster)